# Explaining Predictive Uncertainty with Information Theoretic Shapley Values

**David S. Watson**
King's College London
david.watson@kcl.ac.uk

**Joshua O'Hara**
King's College London

**Niek Tax**
Meta, Central Applied Science

**Richard Mudd**
Meta, Central Applied Science

**Ido Guy**
Meta, Central Applied Science

## Abstract

Researchers in explainable artificial intelligence have developed numerous methods for helping users understand the predictions of complex supervised learning models. By contrast, explaining the *uncertainty* of model outputs has received relatively little attention. We adapt the popular Shapley value framework to explain various types of predictive uncertainty, quantifying each feature's contribution to the conditional entropy of individual model outputs. We consider games with modified characteristic functions and find deep connections between the resulting Shapley values and fundamental quantities from information theory and conditional independence testing. We outline inference procedures for finite sample error rate control with provable guarantees, and implement efficient algorithms that perform well in a range of experiments on real and simulated data. Our method has applications to covariate shift detection, active learning, feature selection, and active feature-value acquisition.

## 1 Introduction

Machine learning (ML) algorithms can solve many prediction tasks with greater accuracy than classical methods. However, some of the most popular and successful algorithms, such as deep neural networks, often produce models with millions of parameters and complex nonlinearities. The resulting "black box" is essentially unintelligible to humans. Researchers in explainable artificial intelligence (XAI) have developed numerous methods to help users better understand the inner workings of such models (see Sect. 2).

Despite the rapid proliferation of XAI tools, the goals of the field have thus far been somewhat narrow. The vast majority of methods in use today aim to explain model *predictions*, i.e. point estimates. But these are not necessarily the only model output of interest. Predictive *uncertainty* can also vary widely across the feature space, in ways that may have a major impact on model performance and human decision making. Such variation makes it risky to rely on the advice of a black box, especially when generalizing to new environments. Discovering the source of uncertainty can be an important first step toward reducing it.

Quantifying predictive uncertainty has many applications in ML. For instance, it is an essential subroutine in any task that involves exploration, e.g. active learning [18, 36], multi-armed bandits [73, 40], and reinforcement learning more generally [55, 76]. Other applications of predictive

37th Conference on Neural Information Processing Systems (NeurIPS 2023).

uncertainty quantification include detecting covariate shift [77] and adversarial examples [70], as well as classification with reject option [23]. Our method aims to expand the scope of XAI to these varied domains by explaining predictive uncertainty via feature attributions.

Knowing the impact of individual features on local uncertainty can help drive data collection and model design. It can be used to detect the source of a suspected covariate shift, select informative features, and test for heteroskedasticity. Our attribution strategy makes use of the Shapley value framework for XAI [83, 44, 74, 49, 9], a popular approach inspired by cooperative game theory, which we adapt by altering the characteristic function and augment with inference procedures for provable error rate control. The approach is fully model-agnostic and therefore not limited to any particular function class.

Our main contributions are threefold: (1) We describe modified variants of the Shapley value algorithm that can explain higher moments of the predictive distribution, thereby extending its explanatory utility beyond mere point estimates. We provide an information theoretic interpretation of the resulting measures and study their properties. (2) We introduce a split conformal inference procedure for Shapley variables with finite sample coverage guarantees. This allows users to test the extent to which attributions for a given feature are concentrated around zero with fixed type I error control. (3) We implement model-specific and model-agnostic variants of our method and illustrate their performance in a range of simulated and real-world experiments, with applications to feature selection, covariate shift detection, and active learning.

## 2 Related Work

XAI has become a major subfield of machine learning in recent years. The focus to date has overwhelmingly been on explaining predictions in supervised learning tasks, most prominently via feature attributions [64, 44, 75], rule lists [65, 38, 71], and counterfactuals [84, 50, 34]. Despite obvious differences between these methods, all arguably share the same goal of identifying minimal conditions sufficient to alter predictions in some pre-specified way [87]. Resulting explanations should be accurate, simple, and relevant for the inquiring agent [86].

Quantifying inductive uncertainty is a fundamental problem in probability theory and statistics, although machine learning poses new challenges and opportunities in this regard [30]. The classical literature on this topic comes primarily from Bayesian modeling [21] and information theory [13], which provide a range of methods for analyzing the distribution of random variables. More recent work on conformal inference [82, 41, 4] has expanded the toolkit for practitioners.

Important application areas for these methods include active learning (AL) and covariate shift detection. In AL, the goal is to selectively query labels for unlabeled instances aimed to maximize classifier improvement under a given query budget. Methods often select instances on which the model has high epistemic (as opposed to aleatoric) uncertainty [67], as for example in BatchBALD [36]. This is especially valuable when labels are sparse and costly to collect while unlabeled data is widely available. In covariate shift detection, the goal is to identify samples that are abnormal relative to the in-distribution observations that the classifier has seen during training. It is well-known that neural networks can be overconfident [22], yielding predictions with unjustifiably high levels of certainty on test samples. Addressing this issue is an active area of research, and a variety of articles take a perspective of quantifying epistemic uncertainty [30, 56]. In safety-critical applications, the degree of model uncertainty can be factored into the decision making, for example by abstaining from prediction altogether when confidence is sufficiently low [23].

Very little work has been done on explaining predictive uncertainty. A notable exception is the CLUE algorithm [3, 42], a model-specific method designed for Bayesian deep learning, which generates counterfactual samples that are maximally similar to some target observation but optimized for minimal conditional variance. This contrasts with our feature attribution approach, which is model-agnostic and thereby the first to make uncertainty explanations available to function classes beyond Bayesian deep learning models.

Predictive uncertainty is often correlated with prediction loss, and therefore explanations of model errors are close relatives of our method. LossSHAP [45] is an extension of Lundberg and Lee [44]'s SHAP algorithm designed to explain the pointwise loss of a supervised learner (e.g., squared error or cross entropy). Though this could plausibly help identify regions where the model is least certain

about predictions, it requires a large labelled test dataset, which may not be available in practice. By contrast, our method only assumes access to some unlabelled dataset of test samples, which is especially valuable when labels are slow or expensive to collect. For instance, LossSHAP is little help in learning environments where covariate shift is detectable before labels are known [94]. This is common, for example, in online advertising [37], where an impression today may lead to a conversion next week but quick detection (and explanation) of covariate shift is vital.

Previous authors have explored information theoretic interpretations of variable importance measures; see [16, Sect. 8.3] for a summary. These methods often operate at global resolutions—e.g., Sobol' indices [57] and SAGE [15]—whereas our focus is on local explanations. Alternatives such as INVASE [90] must be trained alongside the supervised learner itself and are therefore not model-agnostic. L2X [10], REAL-X [32], and SHAP-KL [33] provide post-hoc local explanations, but they require surrogate models to approximate a joint distribution over the full feature space. Chen et al. [11] propose an information theoretic variant of Shapley values for graph-structured data, which we examine more closely in Sect. 4.

## 3  Background

**Notation.**  We use uppercase letters to denote random variables (e.g., $X$) and lowercase for their values (e.g., $x$). Matrices and sets of random variables are denoted by uppercase boldface type (e.g., $\mathbf{X}$) and vectors by lowercase boldface (e.g., $\mathbf{x}$). We occasionally use superscripts to denote samples, e.g. $\mathbf{x}^{(i)}$ is the $i^{\text{th}}$ row of $\mathbf{X}$. Subscripts index features or subsets thereof, e.g. $\mathbf{X}_S = \{X_j\}_{j \in S}$ and $\mathbf{x}_S^{(i)} = \{x_j^{(i)}\}_{j \in S}$, where $S \subseteq [d] = \{1, \ldots, d\}$. We define the complementary subset $\overline{S} = [d] \backslash S$.

**Information Theory.**  Let $p, q$ be two probability distributions over the same $\sigma$-algebra of events. Further, let $p, q$ be absolutely continuous with respect to some appropriate measure. We make use of several fundamental quantities from information theory [13], such as entropy $H(p)$, cross entropy $H(p, q)$, KL-divergence $D_{KL}(p \parallel q)$, and mutual information $I(X; Y)$ (all formally defined in Appx. B.1). We use shorthand for the (conditional) probability mass/density function of the random variable $Y$, e.g. $p_{Y|\mathbf{x}_S} := p(Y \mid \mathbf{X}_S = \mathbf{x}_S)$. We speak interchangeably of the entropy of a random variable and the entropy of the associated mass/density function: $H(Y \mid x) = H(p_{Y|x})$. We call this the *local* conditional entropy to distinguish it from its global counterpart, $H(Y \mid X)$, which requires marginalization over the joint space $\mathcal{X} \times \mathcal{Y}$.

**Shapley Values.**  Consider a supervised learning model $f$ trained on features $\mathbf{X} \in \mathcal{X} \subseteq \mathbb{R}^d$ to predict outcomes $Y \in \mathcal{Y} \subseteq \mathbb{R}$. We assume that data are distributed according to some fixed but unknown distribution $\mathcal{D}$. Shapley values are a feature attribution method in which model predictions are decomposed as a sum: $f(\mathbf{x}) = \phi_0 + \sum_{j=1}^d \phi(j, \mathbf{x})$, where $\phi_0$ is the baseline expectation (i.e., $\phi_0 = \mathbb{E}_{\mathcal{D}}[f(\mathbf{x})]$) and $\phi(j, \mathbf{x})$ denotes the Shapley value of feature $j$ at point $\mathbf{x}$. To define this quantity, we require a value function $v : 2^{[d]} \times \mathbb{R}^d \mapsto \mathbb{R}$ that quantifies the payoff associated with subsets $S \subseteq [d]$ for a particular sample. This characterizes a cooperative game, in which each feature acts as a player. A common choice for defining payoffs in XAI is the following [83, 44, 74, 49, 9]:

$$v_0(S, \mathbf{x}) := \mathbb{E}_{\mathcal{D}}\big[f(\mathbf{x}) \mid \mathbf{X}_S = \mathbf{x}_S\big],$$

where we marginalize over the complementary features $\overline{S}$ in accordance with reference distribution $\mathcal{D}$. For any value function $v$, we may define the following random variable to represent $j$'s marginal contribution to coalition $S$ at point $\mathbf{x}$:

$$\Delta_v(S, j, \mathbf{x}) := v(S \cup \{j\}, \mathbf{x}) - v(S, \mathbf{x}).$$

Then $j$'s Shapley value is just the weighted mean of this variable over all subsets:

$$\phi_v(j, \mathbf{x}) := \sum_{S \subseteq [d] \backslash \{j\}} \frac{|S|! \, (d - |S| - 1)!}{d!} \big[\Delta_v(S, j, \mathbf{x})\big]. \tag{1}$$

It is well known that Eq. 1 is the unique solution to the attribution problem that satisfies certain desirable properties, including efficiency, symmetry, sensitivity, and linearity [74] (for formal statements of these axioms, see Appx. B.2.)

## 4   Alternative Value Functions

To see how standard Shapley values can fall short, consider a simple data generating process with $X, Z \sim \mathcal{U}(0,1)^2$ and $Y \sim \mathcal{N}(X, Z^2)$. Since the true conditional expectation of $Y$ is $X$, this feature will get 100% of the attributions in a game with payoffs given by $v_0$. However, just because $Z$ receives zero attribution does not mean that it adds no information to our predictions—on the contrary, we can use $Z$ to infer the predictive variance of $Y$ and calibrate confidence intervals accordingly. This sort of higher order information is lost in the vast majority of XAI methods.

We consider information theoretic games that assign nonzero attribution to $Z$ in the example above, and study the properties of resulting Shapley values. We start in an idealized scenario in which we have: (i) oracle knowledge of the joint distribution $\mathcal{D}$; and (ii) unlimited computational budget, thereby allowing complete enumeration of all feature subsets.

INVASE [90] is a method for learning a relatively small but maximally informative subset of features $S \subset [d]$ using the loss function $D_{KL}(p_{Y|\mathbf{x}} \parallel p_{Y|\mathbf{x}_S}) + \lambda|S|$, where $\lambda$ is a regularization penalty. Jethani et al. [33]'s SHAP-KL adapts this loss function to define a new game:

$$v_{KL}(S, \mathbf{x}) := -D_{KL}(p_{Y|\mathbf{x}} \parallel p_{Y|\mathbf{x}_S}),$$

which can be interpreted as $-1$ times the excess number of bits one would need on average to describe samples from $Y \mid \mathbf{x}$ given code optimized for $Y \mid \mathbf{x}_S$.

Chen et al. [11] make a similar proposal, replacing KL-divergence with cross entropy:

$$v_{CE}(S, \mathbf{x}) := -H(p_{Y|\mathbf{x}}, p_{Y|\mathbf{x}_S}).$$

This value function is closely related to that of LossSHAP [45], which for likelihood-based loss functions can be written:

$$v_L(S, \mathbf{x}) := -\log p(Y = y \mid \mathbf{x}_S),$$

where $y$ denotes the true value of $Y$ at the point $\mathbf{x}$. As Covert et al. [16] point out, this is equivalent to the pointwise mutual information $I(y; \mathbf{x}_S)$, up to an additive constant. However, $v_L$ requires true labels for $Y$, which may not be available when evaluating feature attributions on a test set. By contrast, $v_{CE}$ averages over $\mathcal{Y}$, thereby avoiding this issue: $v_{CE}(S, \mathbf{x}) = -\mathbb{E}_{Y|\mathbf{x}}\big[v_L(S, \mathbf{x})\big]$. We reiterate that in all cases we condition on some fixed value of $\mathbf{x}$ and do not marginalize over the feature space $\mathcal{X}$. This contrasts with global feature attribution methods like SAGE [15], which can be characterized by averaging $v_L$ over the complete joint distribution $p(\mathbf{X}, Y)$.

It is evident from the definitions that $v_{KL}$ and $v_{CE}$ are equivalent up to an additive constant not depending on $S$, namely $H(p_{Y|\mathbf{x}})$. This renders the resulting Shapley values from both games identical (all proofs in Appx. A.)[1]

**Proposition 4.1.** *For all features $j \in [d]$, coalitions $S \subseteq [d]\setminus\{j\}$, and samples $\mathbf{x} \sim \mathcal{D}_X$:*

$$\Delta_{KL}(S, j, \mathbf{x}) = \Delta_{CE}(S, j, \mathbf{x})$$
$$= \int_{\mathcal{Y}} p(y \mid \mathbf{x}) \log \frac{p(y \mid \mathbf{x}_S, x_j)}{p(y \mid \mathbf{x}_S)} \, dy.$$

This quantity answers the question: if the target distribution were $p_{Y|\mathbf{x}}$, how many more bits of information would we get on average by adding $x_j$ to the conditioning event $\mathbf{x}_S$? Resulting Shapley values summarize each feature's contribution in bits to the distance between $Y$'s fully specified local posterior distribution $p(Y \mid \mathbf{x})$ and the prior $p(Y)$.

**Proposition 4.2.** *With $v \in \{v_{KL}, v_{CE}\}$, Shapley values satisfy $\sum_{j=1}^{d} \phi_v(j, \mathbf{x}) = D_{KL}(p_{Y|\mathbf{x}} \parallel p_Y)$.*

We introduce two novel information theoretic games, characterized by negative and positive local conditional entropies:

$$v_{IG}(S, \mathbf{x}) := -H(Y \mid \mathbf{x}_S), \quad v_H(S, \mathbf{x}) := H(Y \mid \mathbf{x}_S).$$

---

[1]All propositions in this section can be adapted to the classification setting by replacing the integral with a summation over labels $\mathcal{Y}$.

The former subscript stands for information gain; the latter for entropy. Much like $v_{CE}$, these value functions can be understood as weighted averages of LossSHAP payoffs over $\mathcal{Y}$, however this time with expectation over a slightly different distribution: $v_{IG}(S, \mathbf{x}) = -v_H(S, \mathbf{x}) = -\mathbb{E}_{Y|\mathbf{x}_S}[v_L(S, \mathbf{x})]$. The marginal contribution of feature $j$ to coalition $S$ is measured in bits of local conditional mutual information added or lost, respectively (note that $\Delta_{IG} = -\Delta_H$).

**Proposition 4.3.** *For all features $j \in [d]$, coalitions $S \subseteq [d] \setminus \{j\}$, and samples $\mathbf{x} \sim \mathcal{D}_X$:*

$$\Delta_{IG}(S, j, \mathbf{x}) = I(Y; x_j \mid \mathbf{x}_S).$$

This represents the decrease in $Y$'s uncertainty attributable to the conditioning event $x_j$ when we already know $\mathbf{x}_S$. This quantity is similar (but not quite equivalent) to the *information gain*, a common optimization objective in tree growing algorithms [61, 62]. The difference again lies in the fact that we do not marginalize over $\mathcal{X}$, but instead condition on a single instance. Resulting Shapley values summarize each feature's contribution in bits to the overall local information gain.

**Proposition 4.4.** *Under $v_{IG}$, Shapley values satisfy $\sum_{j=1}^d \phi_{IG}(j, \mathbf{x}) = I(Y; \mathbf{x})$.*

In the classic game $v_0$, out-of-coalition features are eliminated by marginalization. However, this will not generally work in our information theoretic games. Consider the modified entropy game, designed to take $d$-dimensional input:

$$v_{H^*}(S, \mathbf{x}) := \mathbb{E}_{\mathcal{D}}\big[H(p_{Y|\mathbf{x}}) \mid \mathbf{X}_S = \mathbf{x}_S\big].$$

This game is not equivalent to $v_H$, as shown in the following proposition.

**Proposition 4.5.** *For all coalitions $S \subset [d]$ and samples $\mathbf{x} \sim \mathcal{D}_X$:*

$$v_H(S, \mathbf{x}) - v_{H^*}(S, \mathbf{x}) = D_{KL}(p_{Y|\mathbf{X}_{\overline{S}}, \mathbf{x}_S} \| p_{Y|\mathbf{x}_S}).$$

The two value functions will tend to diverge when out-of-coalition features $\mathbf{X}_{\overline{S}}$ inform our predictions about $Y$, given prior knowledge of $\mathbf{x}_S$. Resulting Shapley values represent the difference in bits between the local and global conditional entropy.

**Proposition 4.6.** *Under $v_{H^*}$, Shapley values satisfy $\sum_{j=1}^d \phi_{H^*}(j, \mathbf{x}) = H(Y \mid \mathbf{x}) - H(Y \mid \mathbf{X})$.*

In other words, $\phi_{H^*}(j, \mathbf{x})$ is $j$'s contribution to conditional entropy at a given point, compared to a global baseline that averages over all points.

These games share an important and complex relationship to conditional independence structures. We distinguish here between global claims of conditional independence, e.g. $Y \perp\!\!\!\perp X \mid Z$, and local or context-specific independence (CSI), e.g. $Y \perp\!\!\!\perp X \mid z$. The latter occurs when $X$ adds no information about $Y$ under the conditioning event $Z = z$ [7] (see Appx. B.1 for an example).

**Theorem 4.7.** *For value functions $v \in \{v_{KL}, v_{CE}, v_{IG}, v_H\}$, we have:*

*(a) $Y \perp\!\!\!\perp X_j \mid \mathbf{X}_S \Leftrightarrow \sup_{\mathbf{x} \in \mathcal{X}} |\Delta_v(S, j, \mathbf{x})| = 0$.*
*(b) $Y \perp\!\!\!\perp X_j \mid \mathbf{x}_S \Rightarrow \Delta_v(S, j, \mathbf{x}) = 0$.*
*(c) The set of distributions such that $\Delta_v(S, j, \mathbf{x}) = 0 \wedge Y \not\perp\!\!\!\perp X_j \mid \mathbf{x}_S$ is Lebesgue measure zero.*

Item (a) states that $Y$ is conditionally independent of $X_j$ given $\mathbf{X}_S$ if and only if $j$ makes no contribution to $S$ at any point $\mathbf{x}$. Item (b) states that the weaker condition of CSI is sufficient for zero marginal payout. However, while the converse does not hold in general, item (c) states that the set of counterexamples is *small* in a precise sense—namely, it has Lebesgue measure zero. A similar result holds for so-called *unfaithful* distributions in causality [72, 92], in which positive and negative effects cancel out exactly, making it impossible to detect certain graphical structures. Similarly, context-specific dependencies may be obscured when positive and negative log likelihood ratios cancel out as we marginalize over $\mathcal{Y}$. Measure zero events are not necessarily harmless, especially when working with finite samples. Near violations may in fact be quite common due to statistical noise [80]. Together, these results establish a powerful, somewhat subtle link between conditional independencies and information theoretic Shapley values. Similar results are lacking for the standard value function $v_0$—with the notable exception that conditional independence implies zero marginal payout [46]—an inevitable byproduct of the failure to account for predictive uncertainty.

# 5 Method

The information theoretic quantities described in the previous section are often challenging to calculate, as they require extensive conditioning and marginalization. Computing some $\mathcal{O}(2^d)$ such quantities per Shapley value, as Eq. 1 requires, quickly becomes infeasible. (See [81] for an in-depth analysis of the time complexity of Shapley value algorithms.) Therefore, we make several simplifying assumptions that strike a balance between computational tractability and error rate control.

First, we require some uncertainty estimator $h : \mathcal{X} \mapsto \mathbb{R}_{\geq 0}$. Alternatively, we could train new estimators for each coalition [88]; however, this can be impractical for large datasets and/or complex function classes. In the previous section, we assumed access to the true data generating process. In practice, we must train on finite samples, often using outputs from the base model $f$. In the regression setting, this may be a conditional variance estimator, as in heteroskedastic error models [35]; in the classification setting, we assume that $f$ outputs a pmf over class labels and write $f_y : \mathbb{R}^d \mapsto [0, 1]$ to denote the predicted probability of class $y \in \mathcal{Y}$. Then predictive entropy is estimated via the plug-in formula $h_t(\mathbf{x}) := -\sum_{y \in \mathcal{Y}} f_y(\mathbf{x}) \log f_y(\mathbf{x})$, where the subscript $t$ stands for *total*.

In many applications, we must decompose total entropy into epistemic and aleatoric components—i.e., uncertainty arising from the model or the data, respectively. We achieve this via ensemble methods, using a set of $B$ basis functions, $\{f^1, \ldots, f^B\}$. These may be decision trees, as in a random forest [68], or subsets of neural network nodes, as in Monte Carlo (MC) dropout [20]. Let $f_y^b(\mathbf{x})$ be the conditional probability estimate for class $y$ given sample $\mathbf{x}$ for the $b^{\text{th}}$ basis function. Then aleatoric uncertainty is given by $h_a(\mathbf{x}) := -\frac{1}{B} \sum_{b=1}^{B} \sum_{y \in \mathcal{Y}} f_y^b(\mathbf{x}) \log f_y^b(\mathbf{x})$. Epistemic uncertainty is simply the difference [28], $h_e(\mathbf{x}) := h_t(\mathbf{x}) - h_a(\mathbf{x})$.[2] Alternative methods may be appropriate for specific function classes, e.g. Gaussian processes [63] or Bayesian deep learning models [51]. We leave the choice of which uncertainty measure to explain up to practitioners. In what follows, we use the generic $h(\mathbf{x})$ to signify whichever estimator is of relevance for a given application.

We are similarly ecumenical regarding reference distributions. This has been the subject of much debate in recent years, with authors variously arguing that $\mathcal{D}$ should be a simple product of marginals [31]; or that the joint distribution should be modeled for proper conditioning and marginalization [1]; or else that structural information should be encoded to quantify causal effects [25]. Each approach makes sense in certain settings [8, 85], so we leave it up to practitioners to decide which is most appropriate for their use case. We stress that information theoretic games inherit all the advantages and disadvantages of these samplers from the conventional XAI setting, and acknowledge that attributions should be interpreted with caution when models are forced to extrapolate to off-manifold data [26]. Previous work has shown that no single sampling method dominates, with performance varying as a function of data type and function class [53]; see [9] for a discussion.

Finally, we adopt standard methods to efficiently sample candidate coalitions. Observe that the distribution on subsets implied by Eq. 1 induces a symmetric pmf on cardinalities $|S| \in \{0, \ldots, d-1\}$ that places exponentially greater weight at the extrema than it does at the center. Thus while there are over 500 billion coalitions at $d = 40$, we can cover 50% of the total weight by sampling just over 0.1% of these subsets (i.e., those with cardinality $\leq 9$ or $\geq 30$). To reach 90% accuracy requires just over half of all coalitions. In fact, under some reasonable conditions, sampling $\Theta(n)$ coalitions is asymptotically optimal, up to a constant factor [88]. We also employ the paired sampling approach of Covert and Lee [14] to reduce variance and speed up convergence still further.

Several authors have proposed inference procedures for Shapley values [88, 69, 93]. These methods could in principle be extended to our revised games. However, existing algorithms are typically either designed for local inference, in which case they are ill-suited to make global claims about feature relevance, or require global value functions upfront, unlike the local games we consider here. As an alternative, we describe a method for aggregating local statistics for global inference. Specifically, we test whether the random variable $\phi(j, \mathbf{x})$ tends to concentrate around zero for a given $j$. We take a conformal approach [82, 41] that provides the following finite sample coverage guarantee.

**Theorem 5.1** (Coverage). *Partition $n$ training samples $\{(\mathbf{x}^{(i)}, y^{(i)})\}_{i=1}^n \sim \mathcal{D}$ into two equal-sized subsets $\mathcal{I}_1, \mathcal{I}_2$ where $\mathcal{I}_1$ is used for model fitting and $\mathcal{I}_2$ for computing Shapley values. Fix*

---

[2]The additive decomposition of total uncertainty into epistemic and aleatoric components has recently been challenged [89]. While alternative formulations are possible, we stick with this traditional view, which is widely used in deep ensembles and related methods for probabilistic ML [39, 56, 60].

*a target level $\alpha \in (0,1)$ and estimate the upper and lower bounds of the Shapley distribution from the empirical quantiles. That is, let $\hat{q}_{lo}$ be the $\ell$th smallest value of $\phi(j, \mathbf{x}^{(i)}), i \in \mathcal{I}_2$, for $\ell = \lceil (n/2+1)(\alpha/2) \rceil$, and let $\hat{q}_{hi}$ be the $u$th smallest value of the same set, for $u = \lceil (n/2+1)(1-\alpha/2) \rceil$. Then for any test sample $\mathbf{x}^{(n+1)} \sim \mathcal{D}$, we have:*

$$\mathbb{P}\big(\phi(j, \mathbf{x}^{(n+1)}) \in [\hat{q}_{lo}, \hat{q}_{hi}]\big) \geq 1 - \alpha.$$

*Moreover, if Shapley values have a continuous joint distribution, then the upper bound on this probability is $1 - \alpha + 2/(n+2)$.*

Note that this is not a *conditional* coverage claim, insomuch as the bounds are fixed for a given $X_j$ and do not vary with other feature values. However, Thm. 5.1 provides a PAC-style guarantee that Shapley values do not exceed a given (absolute) threshold with high probability, or that zero falls within the $(1 - \alpha) \times 100\%$ confidence interval for a given Shapley value. These results can inform decisions about feature selection, since narrow intervals around zero are necessary (but not sufficient) evidence of uninformative predictors. This result is most relevant for tabular or text data, where features have some consistent meaning across samples; it is less applicable to image data, where individual pixels have no stable interpretation over images.

## 6 Experiments

Full details of all datasets and hyperparameters can be found in Appx. C, along with supplemental experiments that did not fit in the main text. Code for all experiments and figures can be found in our dedicated GitHub repository.[3] We use DeepSHAP to sample out-of-coalition feature values in neural network models, and TreeSHAP for boosted ensembles. Alternative samplers are compared in a separate simulation experiment below. Since our goal is to explain predictive *entropy* rather than *information*, we use the value function $v_{H^*}$ throughout, with plug-in estimators for total, epistemic, and/or aleatoric uncertainty.

### 6.1 Supervised Learning Examples

First, we perform a simple proof of concept experiment that illustrates the method's performance on image, text, and tabular data.

**Image Data.** We examine binary classifiers on subsets of the MNIST dataset. Specifically, we train deep convolutional neural nets to distinguish 1 vs. 7, 3 vs. 8, and 4 vs. 9. These digit pairs tend to look similar in many people's handwriting and are often mistaken for one another. We therefore expect relatively high uncertainty in these examples, and use a variant of DeepSHAP to visualize the pixel-wise contributions to predictive entropy, as estimated via MC dropout. We compute attributions for epistemic and aleatoric uncertainty, visually confirming that the former identifies regions of the image that most increase or reduce uncertainty (see Fig. 1A).

Applying our method, we find that epistemic uncertainty is reduced by the upper loop of the 9, as well as by the downward hook on the 7. By contrast, uncertainty is increased by the odd angle of the 8 and its small bottom loop. Aleatoric uncertainty, by contrast, is more mixed across the pixels, reflecting irreducible noise.

**Text Data.** We apply a transformer network to the IMDB dataset, which contains movie reviews for some 50,000 films. This is a sentiment analysis task, with the goal of identifying positive vs. negative reviews. We visualize the contribution of individual words to the uncertainty of particular predictions as calculated using the modified DeepSHAP pipeline, highlighting how some tokens tend to add or remove predictive information.

We report results for two high-entropy examples in Fig. 1B. In the first review, the model appears confused by the sentence "This is not Great Cinema but I was most entertained," which clearly conveys some ambiguity in the reviewer's sentiment. In the second example, the uncertainty comes from several sources including unexpected juxtapositions such as "laughing and crying", as well as "liar liar...you will love this movie."

---

[3] https://github.com/facebookresearch/infoshap.

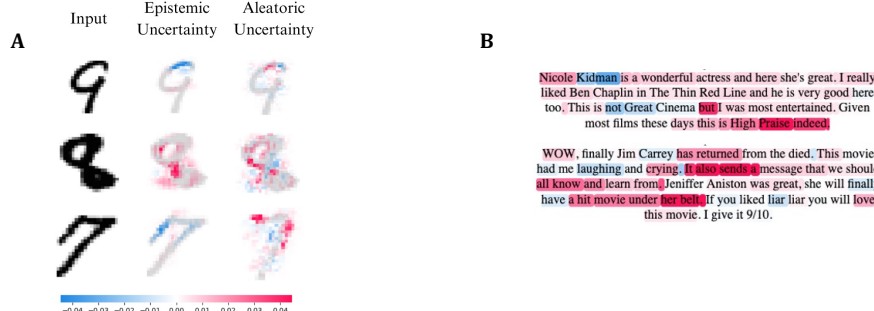

Figure 1: **A**. MNIST examples. We highlight pixels that increase (red) and decrease (blue) predictive uncertainty in digit classification tasks (1 vs. 7, 3 vs. 8, and 4 vs. 9). **B**. Reviews from the IMDB dataset, with tokens colored by their relative contribution to the entropy of sentiment predictions.

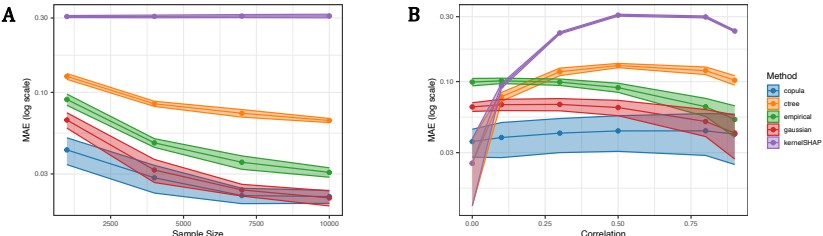

Figure 2: **A**. Mean absolute error (MAE) as a function of sample size, with autocorrelation fixed at $\rho = 0.5$. **B**. MAE as a function of autocorrelation with sample size fixed at $n = 2000$. Shading represents standard errors across 50 replicates.

**Tabular Data.** We design a simple simulation experiment, loosely inspired by [1], in which Shapley values under the entropy game have a closed form solution (see Appx. C.1 for details). This allows us to compare various approaches for sampling out-of-coalition feature values. Variables **X** are multivariate normally distributed with a Toeplitz covariance matrix $\Sigma_{ij} = \rho^{|i-j|}$ and $d = 4$ dimensions. The conditional distribution of outcome variable $Y$ is Gaussian, with mean and variance depending on **x**. We exhaustively enumerate all feature subsets at varying sample sizes and values of the autocorrelation parameter $\rho$.

For imputation schemes, we compare KernelSHAP, maximum likelihood, copula methods, empirical samplers, and ctree (see [54] for definitions of each, and a benchmark study of conditional sampling strategies for feature attributions). We note that this task is strictly more difficult than computing Shapley values under the standard $v_0$, since conditional variance must be estimated from the residuals of a preliminary model, itself estimated from the data. No single method dominates throughout, but most converge on the true Shapley value as sample size increases. Predictably, samplers that take conditional relationships into account tend to do better under autocorrelation than those that do not.

### 6.2 Covariate Shift and Active Learning

To illustrate the utility of our method for explaining covariate shift, we consider several semi-synthetic experiments. We start with four binary classification datasets from the UCI machine learning repository [17]—`BreastCancer`, `Diabetes`, `Ionosphere`, and `Sonar`—and make a random 80/20 train/test split on each. We use an XGBoost model [12] with 50 trees to estimate conditional probabilities and the associated uncertainty. We then perturb a random feature from the test set, adding a small amount of Gaussian noise to alter its underlying distribution. Resulting predictions have a large degree of entropy, and would therefore be ranked highly by an AL acquisition function. We compute information theoretic Shapley values for original and perturbed test sets. Results are visualized in Fig. 3.

Our method clearly identifies the source of uncertainty in these datapoints, assigning large positive or negative attributions to perturbed features in the test environment. Note that the distribution shifts

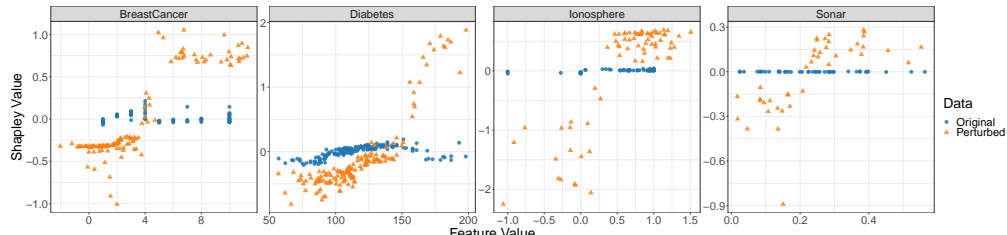

Figure 3: Information theoretic Shapley values explain the uncertainty of predictions on original and perturbed test sets. Our method correctly attributes the excess entropy to the perturbed features.

are fairly subtle in each case, rarely falling outside the support of training values for a given feature. Thus we find that information theoretic Shapley values can be used in conjunction with covariate shift detection algorithms to explain the source of the anomaly, or in conjunction with AL algorithms to explain the exploratory selection procedure.

### 6.3 Feature Selection

Another application of the method is as a feature selection tool when heteroskedasticity is driven by some but not all variables. For this experiment, we modify the classic Friedman benchmark [19], which was originally proposed to test the performance of nonlinear regression methods under signal sparsity. Outcomes are generated according to:

$$Y = 10\sin(\pi X_1 X_2) + 20(X_3 - 0.5)^2 + 10X_4 + 5X_5 + \epsilon_y,$$

with input features $\mathbf{X} \sim \mathcal{U}(0,1)^{10}$ and standard normal residuals $\epsilon_y \sim \mathcal{N}(0,1^2)$. To adapt this DGP to our setting, we scale $Y$ to the unit interval and define:

$$Z = 10\sin(\pi X_6 X_7) + 20(X_8 - 0.5)^2 + 10X_9 + 5X_{10} + \epsilon_z,$$

with $\epsilon_z \sim \mathcal{N}(0, \tilde{Y}^2)$, where $\tilde{Y}$ denotes the rescaled version of $Y$. Note that $Z$'s conditional variance depends exclusively on the first five features, while its conditional mean depends only on the second five. Thus with $f(\mathbf{x}) = \mathbb{E}[Z \mid \mathbf{x}]$ and $h(\mathbf{x}) = \mathbb{V}[Z \mid \mathbf{x}]$, we should expect Shapley values for $f$ to concentrate around zero for $\{X_6, \ldots, X_{10}\}$, while Shapley values for $h$ should do the same for $\{X_1, \ldots, X_5\}$.

We draw 2000 training samples and fit $f$ using XGBoost with 100 trees. This provides estimates of both the conditional mean (via predictions) and the conditional variance (via observed residuals $\hat{\epsilon}_y$). We fit a second XGBoost model $h$ with the same hyperparameters to predict $\log(\hat{\epsilon}_y^2)$. Results are reported on a test set of size 1000. We compute attributions using TreeSHAP [45] and visualize results in Fig. 4A. We find that Shapley values are clustered around zero for unimportant features in each model, demonstrating the method's promise for discriminating between different modes of predictive information. In a supplemental experiment, we empirically evaluate our conformal coverage guarantee on this same task, achieving nominal coverage at $\alpha = 0.1$ for all features (see Appx. C.3).

As an active feature-value acquisition example, we use the same modified Friedman benchmark, but this time increase the training sample size to 5000 and randomly delete some proportion of cells in the design matrix for $\mathbf{X}$. This simulates the effect of missing data, which may arise due to entry errors or high collection costs. XGBoost has native methods for handling missing data at training and test time, although resulting Shapley values are inevitably noisy. We refit the conditional variance estimator $h$ and record feature rankings with variable missingness.

The goal in active feature-value acquisition is to prioritize the variables whose values will best inform future predictions subject to budgetary constraints. Fig. 4B shows receiver operating characteristic (ROC) curves for a feature importance ranking task as the frequency of missing data increases from zero to 50%. Importance is estimated via absolute Shapley values. Though performance degrades with increased missing data, as expected, we find that our method reliably ranks important features above unimportant ones in all trials. Even with fully half the data missing, we find an AUC of 0.682, substantially better than random.

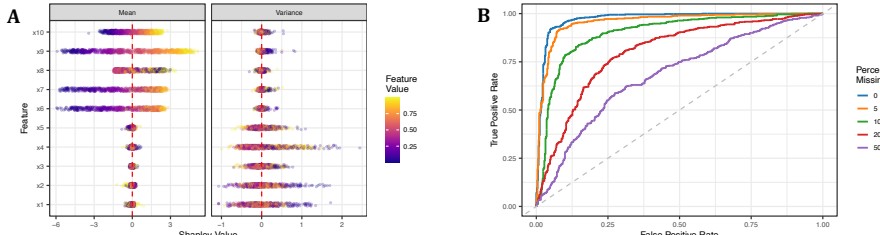

Figure 4: **A**. Results for the modified Friedman benchmark experiment. The conditional mean depends on $\{X_6, \dots, X_{10}\}$, while the conditional variance relies on $\{X_1, \dots, X_5\}$. **B**. ROC curves for a feature ranking task with variable levels of missingness. The proposed value function gives informative results for feature-value acquisition.

## 7 Discussion

Critics have long complained that Shapley values (using the conventional payoff function $v_0$) are difficult to interpret in XAI. It is not always clear what it even means to remove features [43, 2], and large/small attributions are neither necessary nor sufficient for important/unimportant predictors, respectively [5, 29]. In an effort to ground these methods in classical statistical notions, several authors have analyzed Shapley values in the context of ANOVA decompositions [24, 6] or conditional independence tests [47, 78], with mixed results. Our information theoretic approach provides another window into this debate. With modified value functions, we show that marginal payoffs $\Delta_v(S, j, \mathbf{x})$ have an unambiguous interpretation as a local dependence measure. Still, Shapley values muddy the waters somewhat by averaging these payoffs over coalitions.

There has been a great deal of interest in recent years on *functional data analysis*, where the goal is to model not just the conditional mean of the response variable $\mathbb{E}[Y \mid \mathbf{x}]$, but rather the entire distribution $P(Y \mid \mathbf{x})$, including higher moments. Distributional regression techniques have been developed for additive models [66], gradient boosting machines [79], random forests [27], and neural density estimators [58]. Few if any XAI methods have been specifically designed to explain such models, perhaps because attributions would be heavily weighted toward features with a significant impact on the conditional expectation, thereby simply reducing to classic measures. Our method provides one possible way to disentangle those attributions and focus attention on higher moments. Future work will explore more explicit connections to the domain of functional data.

One advantage of our approach is its modularity. We consider a range of different information theoretic games, each characterized by a unique value function. We are agnostic about how to estimate the relevant uncertainty measures, fix reference distributions, or sample candidate coalitions. These are all active areas of research in their own right, and practitioners should choose whichever combination of tools works best for their purpose.

However, this flexibility does not come for free. Computing Shapley values for many common function classes is #P-hard, even when features are jointly independent [81]. Modeling dependencies to impute values for out-of-coalition features is a statistical challenge that requires extensive marginalization. Some speedups can be achieved by making convenient assumptions, but these may incur substantial errors in practice. These are familiar problems in feature attribution tasks. Our method inherits the same benefits and drawbacks.

## 8 Conclusion

We introduced a range of methods to explain conditional entropy in ML models, bringing together existing work on uncertainty quantification and feature attributions. We studied the information theoretic properties of several games, and implemented our approach in model-specific and model-agnostic algorithms with numerous applications. Future work will continue to examine how XAI can go beyond its origins in prediction to inform decision making in areas requiring an exploration-exploitation trade-off, such as bandits and reinforcement learning.

## Acknowledgments and Disclosure of Funding

We thank Matthew Shorvon for his feedback on an earlier draft of this paper. We are also grateful to the reviewers for their helpful comments.

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

# A Proofs

**Proof of Prop. 4.1.** Substituting $v_{KL}$ into the definition of $\Delta_v(S, j, \mathbf{x})$ gives:

$$\Delta_{KL}(S, j, \mathbf{x}) = -D_{KL}(p_{Y|\mathbf{x}} \parallel p_{Y|\mathbf{x}_S, x_j}) + D_{KL}(p_{Y|\mathbf{x}} \parallel p_{Y|\mathbf{x}_S}).$$

Rearranging and using the definition of KL-divergence, we have:

$$\Delta_{KL}(S, j, \mathbf{x}) = \underset{Y|\mathbf{x}}{\mathbb{E}} \big[ \log p(y \mid \mathbf{x}) - \log p(y \mid \mathbf{x}_S) \big] - \underset{Y|\mathbf{x}}{\mathbb{E}} \big[ \log p(y \mid \mathbf{x}) - \log p(y \mid \mathbf{x}_S, x_j) \big].$$

Cleaning up in steps:

$$\Delta_{KL}(S, j, \mathbf{x}) = \underset{Y|\mathbf{x}}{\mathbb{E}} \big[ \log p(y \mid \mathbf{x}) - \log p(y \mid \mathbf{x}_S) - \log p(y \mid \mathbf{x}) + \log p(y \mid \mathbf{x}_S, x_j) \big]$$

$$= \underset{Y|\mathbf{x}}{\mathbb{E}} \big[ \log p(y \mid \mathbf{x}_S, x_j) - \log p(y \mid \mathbf{x}_S) \big]$$

$$= \int_{\mathcal{Y}} p(y \mid \mathbf{x}) \, \log \frac{p(y \mid \mathbf{x}_S, x_j)}{p(y \mid \mathbf{x}_S)} \, dy.$$

Substituting $v_{CE}$ into the definition of $\Delta_v(S, j, \mathbf{x})$ gives:

$$\Delta_{CE}(S, j, \mathbf{x}) = -H(p_{Y|\mathbf{x}}, p_{Y|\mathbf{x}_S, x_j}) + H(p_{Y|\mathbf{x}}, p_{Y|\mathbf{x}_S}).$$

Rearranging and using the definition of cross entropy, we have:

$$\Delta_{CE}(S, j, \mathbf{x}) = H(p_{Y|\mathbf{x}}, p_{Y|\mathbf{x}_S}) - H(p_{Y|\mathbf{x}}, p_{Y|\mathbf{x}_{S \cup \{j\}}})$$

$$= \underset{Y|\mathbf{x}}{\mathbb{E}} \big[ -\log p(y \mid \mathbf{x}_S) \big] - \underset{Y|\mathbf{x}}{\mathbb{E}} \big[ -\log p(y \mid \mathbf{x}_S, x_j) \big]$$

$$= \underset{Y|\mathbf{x}}{\mathbb{E}} \big[ \log p(y \mid \mathbf{x}_S, x_j) - \log p(y \mid \mathbf{x}_S) \big]$$

$$= \int_{\mathcal{Y}} p(y \mid \mathbf{x}) \, \log \frac{p(y \mid \mathbf{x}_S, x_j)}{p(y \mid \mathbf{x}_S)} \, dy.$$

**Proof of Prop. 4.2.** Since the Shapley value $\phi_v(j, \mathbf{x})$ is just the expectation of $\Delta_v(S, j, \mathbf{x})$ under a certain distribution on coalitions $S \subseteq [d]\backslash\{j\}$ (see Eq. 1), it follows from Prop. 4.1 that feature attributions will be identical under $v_{KL}$ and $v_{CE}$. To show that resulting Shapley values sum to the KL-divergence between $p(Y \mid \mathbf{x})$ and $p(Y)$, we exploit the efficiency property:

$$\sum_{j=1}^{d} \phi_{KL}(j, \mathbf{x}) = v_{KL}([d], \mathbf{x}) - v_{KL}(\emptyset, \mathbf{x})$$

$$= -D_{KL}(p_{Y|\mathbf{x}} \parallel p_{Y|\mathbf{x}}) + D_{KL}(p_{Y|\mathbf{x}} \parallel p_Y)$$

$$= D_{KL}(p_{Y|\mathbf{x}} \parallel p_Y).$$

The last step exploits Gibbs's inequality, according to which $D_{KL}(p \parallel q) \geq 0$, with $D_{KL}(p \parallel q) = 0$ iff $p = q$.

**Proof of Prop. 4.3.** Substituting $v_{IG}$ into the definition of $\Delta_v(S, j, \mathbf{x})$ gives:

$$\Delta_{IG}(S, j, \mathbf{x}) = -H(Y \mid \mathbf{x}_S, x_j) + H(Y \mid \mathbf{x}_S)$$

$$= H(Y \mid \mathbf{x}_S) - H(Y \mid \mathbf{x}_S, x_j)$$

$$= I(Y; x_j \mid \mathbf{x}_S)$$

$$= \int_{\mathcal{Y}} p(y, x_j \mid \mathbf{x}_S) \log \frac{p(y, x_j \mid \mathbf{x}_S)}{p(y \mid \mathbf{x}_S) \, p(x_j \mid \mathbf{x}_S)} dy.$$

In the penultimate line, we exploit the equality $I(Y; X) = H(Y) - H(Y \mid X)$, by which we define mutual information (see Appx. B.1).

**Proof of Prop. 4.4.** We once again rely on efficiency and the definition of mutual information in terms of marginal and conditional entropy:

$$\sum_{j=1}^{d} \phi_{IG}(j, \mathbf{x}) = v_{IG}([d], \mathbf{x}) - v_{IG}(\emptyset, \mathbf{x})$$

$$= -H(Y \mid \mathbf{x}) + H(Y)$$

$$= H(Y) - H(Y \mid \mathbf{x})$$

$$= I(Y; \mathbf{x}).$$

**Proof of Prop. 4.5.** Let $b(S, \mathbf{x})$ denote the gap between local conditional entropy formulae that take $|S|$- and $d$-dimensional input, respectively. Then we have:

$$
\begin{aligned}
b(S, \mathbf{x}) &= v_H(S, \mathbf{x}) - v_{H^*}(S, \mathbf{x}) \\
&= H(Y \mid \mathbf{x}_S) - \mathop{\mathbb{E}}_{\mathbf{X}_{\overline{S}} \mid \mathbf{x}_S} \big[ H(Y \mid \mathbf{x}) \mid \mathbf{X}_S = \mathbf{x}_S \big] \\
&= - \mathop{\mathbb{E}}_{Y \mid \mathbf{x}_S} \big[ \log p(y \mid \mathbf{x}_S) \big] + \mathop{\mathbb{E}}_{\mathbf{X}_{\overline{S}}, Y \mid \mathbf{x}_S} \big[ \log p(y \mid \mathbf{x}_{\overline{S}}, \mathbf{x}_S) \big] \\
&= \mathop{\mathbb{E}}_{\mathbf{X}_{\overline{S}}, Y \mid \mathbf{x}_S} \big[ \log p(y \mid \mathbf{x}_{\overline{S}}, \mathbf{x}_S) - \log p(y \mid \mathbf{x}_S) \big] \\
&= \int_{\mathcal{X}_{\overline{S}}} \int_{\mathcal{Y}} p(\mathbf{x}_{\overline{S}}, y \mid \mathbf{x}_S) \log \frac{p(y \mid \mathbf{x}_{\overline{S}}, \mathbf{x}_S)}{p(y \mid \mathbf{x}_S)} \, d\mathbf{x}_{\overline{S}} \, dy \\
&= D_{KL}(p(Y \mid \mathbf{X}_{\overline{S}}, \mathbf{x}_S) \parallel p(Y \mid \mathbf{x}_S)).
\end{aligned}
$$

**Proof of Prop. 4.6.** Exploiting the efficiency property, we immediately have:

$$
\begin{aligned}
\sum_{j=1}^{d} \phi_{H^*}(j, \mathbf{x}) &= v_{H^*}([d], \mathbf{x}) - v_{H^*}(\emptyset, \mathbf{x}) \\
&= H(Y \mid \mathbf{x}) - \mathop{\mathbb{E}}_{\mathcal{D}_X} \big[ H(Y \mid \mathbf{x}) \big] \\
&= H(Y \mid \mathbf{x}) - H(Y \mid \mathbf{X}).
\end{aligned}
$$

By contrast, Shapley values for the original entropy game $v_H$ sum to $H(Y \mid \mathbf{x}) - H(Y) = -I(Y; \mathbf{x})$. Thus whereas the baseline for an empty coalition $S = \emptyset$ is the prior entropy $H(Y)$ under $v_H$, the corresponding baseline for the $d$-dimensional version $v_{H^*}$ is the global posterior entropy $H(Y \mid \mathbf{X})$.

**Proof of Thm. 4.7.** Begin with item (a). Note that the conditional independence statement $Y \perp\!\!\!\perp X_j \mid \mathbf{X}_S$ holds iff, for all points $(\mathbf{x}, y) \sim \mathcal{D}$, we have:

$$
p(y \mid \mathbf{x}_S, x_j) = p(y \mid \mathbf{x}_S) \quad \text{and} \quad p(y, x_j \mid \mathbf{x}_S) = p(y \mid \mathbf{x}_S) \, p(x_j \mid \mathbf{x}_S).
$$

The former guarantees that marginal payouts evaluate to zero for $v \in \{v_{KL}, v_{CE}\}$; the latter does the same for $v \in \{v_{IG}, v_H\}$. This follows because the log ratio in each formula evaluates to zero when numerator and denominator are equal.

Of course, conditional independence is also sufficient for zero marginal payout with more familiar value functions such as $v_0$. But item (a) makes an additional claim—that the *converse* holds as well, i.e. that conditional independence is *necessary* for zero marginal payout across all $\mathbf{x}$. This follows from the definitions of the value functions themselves. Observe:

$$
\begin{aligned}
\mathop{\mathbb{E}}_{\mathbf{x} \sim \mathcal{D}_X} \big[ \Delta_{KL}(S, j, \mathbf{x}) \big] &= \mathop{\mathbb{E}}_{(\mathbf{x}, y) \sim \mathcal{D}} \left[ \log \frac{p(y \mid \mathbf{x}_S, x_j)}{p(y \mid \mathbf{x}_S)} \right] \\
&= \mathop{\mathbb{E}}_{\mathcal{D}_X} \left[ \mathop{\mathbb{E}}_{Y \mid \mathbf{x}_S, x_j} \left[ \log \frac{p(y \mid \mathbf{x}_S, x_j)}{p(y \mid \mathbf{x}_S)} \right] \right] \\
&= \mathop{\mathbb{E}}_{\mathcal{D}_X} \big[ D_{KL}(p_{Y \mid \mathbf{x}_S, x_j} \parallel p_{Y \mid \mathbf{x}_S}) \big]
\end{aligned}
$$

By Gibbs's inequality, the KL-divergence between two distributions is zero iff they are equal, so setting this value to zero for all $\mathbf{x}$ satisfies the first definition of conditional independence above. For the latter, we simply point out that:

$$
\mathop{\mathbb{E}}_{\mathbf{x} \sim \mathcal{D}_X} \big[ \Delta_{IG}(S, j, \mathbf{x}) \big] = I(Y; X_j \mid \mathbf{X}_S).
$$

Since conditional mutual information equals zero iff the relevant variables are conditionally independent, this satisfies the second definition above.

Item (b) states that CSI, which is strictly weaker than standard conditional independence, is also sufficient for zero marginal payout at a given point $\mathbf{x}$. This follows directly from the sufficiency argument above.

The converse relationship is more complex, however. Call a distribution *conspiratorial* if there exists some $S, j, \mathbf{x}$ such that $\Delta_v(S, j, \mathbf{x}) = 0 \wedge Y \not\!\perp\!\!\!\perp X_j \mid \mathbf{x}_S$ for some $v \in \{v_{KL}, v_{CE}, v_{IG}, v_H\}$. Such distributions are so named because the relevant probabilities must coordinate in a very specific way to guarantee summation to zero as we marginalize over $\mathcal{Y}$. As a concrete example, consider the following data generating process:

$$
X \sim \text{Bern}(0.5), \quad Z \sim \text{Bern}(0.5), \quad Y \sim \text{Bern}(0.3 + 0.4X - 0.2Z).
$$

What is the contribution of $X$ to coalition $S = \emptyset$ when $X = 1$ and $Z = 1$? In this case, we have neither global nor context-specific independence, i.e. $Y \not\perp X$. Yet, evaluating the payoffs in a KL-divergence game, we have:

$$
\begin{aligned}
\Delta_{KL}(S, j, \mathbf{x}) &= \sum_y P(y \mid X = 1, Z = 1) \ \log \frac{P(y \mid X = 1)}{P(y)} \\
&= 0.5 \log \frac{0.4}{0.6} + 0.5 \log \frac{0.6}{0.4} \\
&= 0.
\end{aligned}
$$

In this case, we find that negative and positive values of the log ratio cancel out exactly as we marginalize over $\mathcal{Y}$. (Similar examples can be constructed for all our information theoretic games.) This shows that CSI is sufficient but not necessary for $\Delta_v(S, j, \mathbf{x}) = 0$.

However, just because conspiratorial distributions are possible does not mean that they are common. Item (c) states that the set of all such distributions has Lebesgue measure zero. Our proof strategy here follows that of Meek [48], who demonstrates a similar result in the case of *unfaithful* distributions, i.e. those whose (conditional) independencies are not entailed by the data's underlying graphical structure. This is an important topic in the causal discovery literature (see, e.g., [91, 92]).

For simplicity, assume a discrete state space $\mathcal{X} \times \mathcal{Y}$, such that the data generating process is fully parametrized by a table of probabilities $T$ for each fully specified event. Fix some $S, j$ such that $Y \not\perp X_j \mid \mathbf{x_S}$. Let $C$ be the number of possible outcomes, $\mathcal{Y} = \{y_1, \ldots, y_C\}$. Define vectors $\mathbf{p}, \mathbf{q}, \mathbf{r}$ of length $C$ such that, for each $c \in [C]$:

$$
p_c = p(y_c \mid \mathbf{x}), \quad q_c = p(y_c \mid \mathbf{x}_S, x_j) - p(y_c \mid \mathbf{x}_S), \quad r_c = \log \frac{p(y_c \mid \mathbf{x}_S, x_j)}{p(y_c \mid \mathbf{x}_S)},
$$

and stipulate that $p(y_c \mid \mathbf{x}_S) > 0$ for all $c \in [C]$ to avoid division by zero. Observe that these variables are all deterministic functions of the parameters encoded in the probability table $T$. By the assumption of local conditional dependence, we know that $\|\mathbf{r}\|_0 > 0$. Yet for our conspiracy to obtain, the data must satisfy $\mathbf{p} \cdot \mathbf{r} = 0$. A well-known algebraic lemma of Okamoto [52] states that if a polynomial constraint is non-trivial (i.e., if there exists some settings for which it does not hold), then the subset of parameters for which it does hold has Lebesgue measure zero. The log ratio $\mathbf{r}$ is not polynomial in $T$, but the difference $\mathbf{q}$ is. The latter also has a strictly positive $L_0$ norm by the assumption of local conditional independence. Crucially, the resulting dot product intersects with our previous dot product at zero. That is, though differences are in general a non-injective surjective function of log ratios, we have a bijection at the origin whereby $\mathbf{p} \cdot \mathbf{r} = 0 \Leftrightarrow \mathbf{p} \cdot \mathbf{q} = 0$. Thus the conspiracy requires nontrivial constraints that are linear in $\mathbf{p}, \mathbf{q}$—themselves polynomial in the system parameters $T$—so we conclude that the set of conspiratorial distributions has Lebesgue measure zero.

**Proof of Thm. 5.1.** Our proof is an application of the split conformal method (see [41, Thm. 2.2]). Whereas that method was designed to bound the distance between predicted and observed outcomes for a regression task, we adapt the argument to measure the concentration of Shapley values for a given feature. To achieve this, we replace out-of-sample absolute residuals with out-of-sample Shapley values and drop the symmetry assumption, which will not generally apply when features are informative. The result follows immediately from the exchangeability of $\phi(j, \mathbf{x}^{(i+1)})$ and $\phi(j, \mathbf{x}^{(i)}), i \in \mathcal{I}_2$, which is itself a direct implication of the i.i.d. assumption. Since bounds are calculated so as to cover $(1 - \alpha) \times 100\%$ of the distribution, it is unlikely that new samples will fall outside this region. Specifically, such exceptions occur with probability at most $\alpha$. This amounts to a sort of PAC guarantee, i.e. that Shapley values will be within a data-dependent interval with probability at least $1 - \alpha$. The interval can be trivially inverted to compute an associated $p$-value.

We identify three potential sources of error in estimating upper and lower quantiles of the Shapley distribution: (1) learning the conditional entropy model $h$ from finite data; (2) sampling values for out-of-coalition features $\overline{S}$; (3) sampling coalitions $S$. Convergence rates as a function of (1) and (2) are entirely dependent on the selected subroutines. With consistent methods for both, conformal prediction bands are provably close to the oracle band (see [41, Thm. 8]). As for (3), Williamson and Feng [88] show that with efficient estimators for (1) and (2), as well as an extra condition on the minimum number of subsets, sampling $\Theta(n)$ coalitions is asymptotically optimal, up to a constant factor.

# B Addenda

This section includes extra background material on information theory and Shapley values.

## B.1 Information Theory

Let $p, q$ be two probability distributions over the same $\sigma$-algebra of events. Further, let $p, q$ be absolutely continuous with respect to some appropriate measure. The *entropy* of $p$ is defined as $H(p) := \mathbb{E}_p[-\log p]$,

i.e. the expected number of bits required to encode the distribution.[4] The *cross entropy* of $p$ and $q$ is defined as $H(p, q) := \mathbb{E}_p[-\log q]$, i.e. the expected number of bits required to encode samples from $p$ using code optimized for $q$. The *KL-divergence* between $p$ and $q$ is defined as $D_{KL}(p \parallel q) := \mathbb{E}_p[\log p/q]$, i.e. the cost in bits of modeling $p$ with $q$. These three quantities are related by the formula $D_{KL}(p \parallel q) = H(p, q) - H(p)$. The reduction in $Y$'s uncertainty attributable to $X$ is also called the *mutual information*, $I(Y; X) := H(Y) - H(Y \mid X)$. This quantity is nonnegative, with $I(Y; X) = 0$ if and only if the variables are independent.

However, conditioning on a specific value of $X$ may increase uncertainty in $Y$, in which case the local posterior entropy exceeds the prior. Thus it is possible that $H(Y \mid x) > H(Y)$ for some $x \in \mathcal{X}$. For example, consider the following data generating process:

$$X \sim \text{Bern}(0.8), \quad Y \sim \text{Bern}(0.5 + 0.25X).$$

In this case, we have $P(Y = 1) = 0.7, P(Y = 1 \mid X = 0) = 0.5$, and $P(Y = 1 \mid X = 1) = 0.75$. It is easy to see that even though the marginal entropy $H(Y)$ exceeds the global conditional entropy $H(Y \mid X)$, the local entropy at $X = 0$ is larger than either quantity, $H(Y \mid X = 0) > H(Y) > H(Y \mid X)$. In other words, conditioning on the event $X = 0$ increases our uncertainty about $Y$.

Similarly, there may be cases in which $I(Y; X \mid Z) > 0$, but $I(Y; X \mid z) = 0$. This is what Boutilier et al. [7] call *context-specific independence* (CSI). For instance, if $X, Z \in \{0, 1\}^2$ and $Y := \max\{X, Z\}$, then we have $Y \not\perp\!\!\!\perp X \mid Z$, but $Y \perp\!\!\!\perp X \mid (Z = 1)$ since $Y$'s value is determined as soon as we know that either parent is 1.

## B.2 The Shapley Axioms

For completeness, we here list the Shapley axioms.

**Efficiency.** Shapley values sum to the difference in payoff between complete and null coalitions:

$$\sum_{j=1}^{d} \phi(j, \mathbf{x}) = v([d], \mathbf{x}) - v(\emptyset, \mathbf{x}).$$

**Symmetry.** If two players make identical contributions to all coalitions, then their Shapley values are equal:

$$\forall S \subseteq [d] \backslash \{i, j\} : v(S \cup \{i\}, \mathbf{x}) = v(S \cup \{j\}, \mathbf{x}) \Rightarrow \phi(i, \mathbf{x}) = \phi(j, \mathbf{x}).$$

**Sensitivity.** If a player makes zero contribution to all coalitions, then its Shapley value is zero:

$$\forall S \subseteq [d] \backslash \{j\} : v(S \cup \{j\}, \mathbf{x}) = v(S, \mathbf{x}) \Rightarrow \phi(j, \mathbf{x}) = 0.$$

**Linearity.** The Shapley value for a convex combination of games can be decomposed into a convex combination of Shapley values. For any $a, b \in \mathbb{R}$ and value functions $v_1, v_2$, we have:

$$\phi_{a \cdot v_1 + b \cdot v_2}(j, \mathbf{x}) = a\phi_{v_1}(j, \mathbf{x}) + b\phi_{v_2}(j, \mathbf{x}).$$

# C Experiments

## C.1 Datasets.

The MNIST dataset is available online.[5] The IMDB dataset is available on Kaggle.[6] For the tabular data experiment in Sect. 6.1, we generate $Y$ according to the following process:

$$\mu(\mathbf{x}) := \boldsymbol{\beta}^\top \mathbf{x}, \quad \sigma^2(\mathbf{x}) := \exp(\boldsymbol{\gamma}^\top \mathbf{x}),$$
$$Y \mid \mathbf{x} \sim \mathcal{N}(\mu(\mathbf{x}), \sigma^2(\mathbf{x})).$$

Coefficients $\boldsymbol{\beta}, \boldsymbol{\gamma}$ are independent Rademacher distributed random vectors of length 4.

The `BreastCancer`, `Diabetes`, `Ionosphere`, and `Sonar` datasets are all distributed in the `mlbench` package, which is available on `CRAN`.[7]

---

[4]Though the term "bit" is technically reserved for units of information measured with logarithmic base 2, we use the word somewhat more loosely to refer to any unit of information.

[5]http://yann.lecun.com/exdb/mnist/.

[6]https://www.kaggle.com/datasets/lakshmi25npathi/imdb-dataset-of-50k-movie-reviews.

[7]https://cran.r-project.org/web/packages/mlbench/index.html.

Table 1: Estimated quantiles and nominal coverage at $\alpha = 0.1$ for Shapley values from the conditional mean and conditional variance models. Results are averaged over 50 replicates.

| Feature | Mean | | | Variance | | |
|---|---|---|---|---|---|---|
| | $\hat{q}_{\text{lo}}$ | $\hat{q}_{\text{hi}}$ | Coverage | $\hat{q}_{\text{lo}}$ | $\hat{q}_{\text{hi}}$ | Coverage |
| $X_1$ | -0.150 | 0.070 | 0.898 | -0.512 | 0.541 | 0.900 |
| $X_2$ | -0.138 | 0.127 | 0.897 | -0.409 | 0.588 | 0.900 |
| $X_3$ | -0.070 | 0.091 | 0.904 | -0.227 | 0.302 | 0.900 |
| $X_4$ | -0.094 | 0.096 | 0.900 | -0.477 | 1.025 | 0.898 |
| $X_5$ | -0.149 | 0.093 | 0.904 | -0.266 | 0.324 | 0.901 |
| $X_6$ | -4.310 | 2.203 | 0.895 | -0.041 | 0.057 | 0.904 |
| $X_7$ | -4.496 | 2.268 | 0.897 | -0.047 | 0.103 | 0.902 |
| $X_8$ | -1.279 | 2.293 | 0.898 | -0.047 | 0.051 | 0.897 |
| $X_9$ | -4.588 | 4.134 | 0.900 | -0.052 | 0.129 | 0.900 |
| $X_{10}$ | -1.886 | 1.927 | 0.898 | -0.051 | 0.065 | 0.902 |

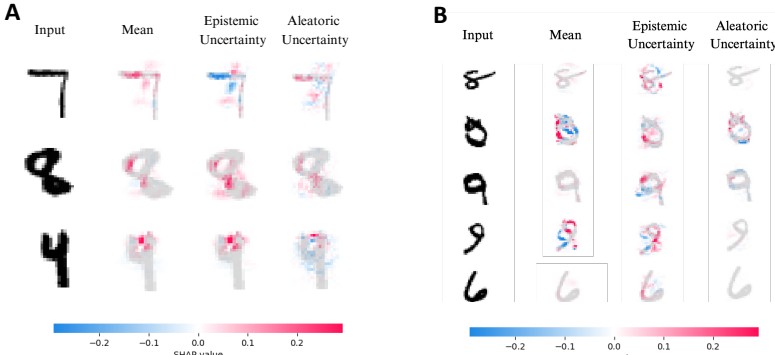

Figure 5: Binary MNIST figure (A) and multiclass MNIST (B), featuring a "Mean" column for standard SHAP outputs. For this column, red/blue pixels indicate features that increase/decrease the probability of the true class label. These values are anticorrelated with epistemic uncertainty, as expected. The effect is more salient in the binary setting, since probability mass is spread out over 10 classes in the multiclass model.

## C.2  Models.

All neural network training was conducted in PyTorch [59]. For the MNIST experiment, we train a deep neural network with the following model architecture: (1) A convolutional layer with 10 filters of size $5 \times 5$, followed by max pooling of size $2 \times 2$, ReLU activation, and a dropout layer with probability 0.3. (2) A convolutional layer with 20 filters of size $5 \times 5$, followed by a dropout layer, max pooling of size $2 \times 2$, ReLU activation, and a dropout layer with probability 0.3. (3) Fully connected (dense) layer with 320 input features and 50 output units, followed by ReLU activation and a dropout layer. (4) Fully connected layer with 50 input features and 10 output units, followed by softmax activation. We train with a batch size of 128 for 20 epochs at a learning rate of 0.01 and momentum 0.5. For Monte Carlo dropout, we do 50 forward passes to sample $B = 50$ subnetworks. For the IMDB experiment, we use a pre-trained BERT model from the Hugging Face transformers library.[8] All hyperparameters are set to their default values. All XGBoost models are trained with the default hyperparameters, with the number of training rounds cited in the text.

## C.3  Coverage

To empirically test our conformal coverage guarantee, we compute quantiles for out-of-sample Shapley values on the modified Friedman benchmark. Results for conditional expectation and conditional variance are reported in Table 1, with target level $\alpha = 0.1$. Note that what constitutes a "small" or "large" interval is context-dependent. The conditional variance model is fit to $\epsilon_y^2$, which has a tighter range than $Z$, leading to smaller Shapley values on average. However, nominal coverage is very close to the target 90% throughout, illustrating how the conformal method can be used for feature selection and outlier detection.

---

[8]https://huggingface.co/docs/transformers/model_doc/bert.

## C.4 Extended MNIST

We supplement the original MNIST experiment with extra examples, including a mean column for the true label (Fig. 5A) and augmenting the binary problem with the classic 10-class task (Fig. 5B). We find that mean and entropy are highly associated in the binary setting, as the analytic formulae suggest, while the relationship is somewhat more opaque in the 10-class setting, since we must average over a larger set of signals.

