# OpenReview forum: "Explaining Predictive Uncertainty with Information Theoretic Shapley Values"
_NeurIPS.cc/2023/Conference — NeurIPS 2023 poster_

### Official Review · Reviewer_TWY9 · 2023-07-03

**Soundness:** 2 fair
**Presentation:** 2 fair
**Contribution:** 2 fair
**Rating:** 4
**Confidence:** 5

**Summary:**

The paper aims to define Shapley values to decompose the conditional distribution of the outcome so to attribute the uncertainty of the outcome to individual variables. This aims to extend prior work, which has mainly focused on explaining the change in the conditional mean of the outcome.

**Strengths:**

The manuscript highlight how methods for explaining the conditional variance can have highly different results from those for explaining the conditional mean. This is nicely shown in the simulation results in Section 6.3 and Figure 3. The authors also nicely highlight how methods for explaining the conditional variance can be useful in many contexts, such as for model calibration.

**Weaknesses:**

1. There are a number of other papers that have studied how to explain the conditional variance of Y given X using Shapley values. For instance, Williamson and Feng 2020 use Shapley values to determine how well individual features predict Y using some user-defined scoring function V. If this scoring function is set to the deviance/KL divergence, then the resulting Shapley values are highly similar to those discussed in this paper. Moreover, this paper discusses how to efficiently perform statistical inference that sidestep computational issues that arise when considering every possible variable subset. The authors should put the current manuscript in context, given such existing works.

2. The link between conditional independencies and the information theoretic Shapley values defined in this paper follows very directly from how the Shapley values were defined. I don't see why the authors state that this connection is "deep" or "subtle". Perhaps the authors can clarify why this connection is far from obvious.

3. The proposed Shapley value is characterizing properties of the oracle distribution. Given the framing of the paper, the true target for statistical inference should be the Shapley value with respect to the oracle distribution. However, the result in Theorem 5.1 does not account for the uncertainty in estimating the oracle distribution and performs inference for a different target (seems to be the mean of the Shapley value of the estimated distribution). Moreover, it is not clear to me what the use case is for marginal coverage guarantees like that presented in Theorem 5.1. (Conditional coverage guarantees are ideal, but I agree that is unattainable.) I suggest better motivating (i) why are we doing statistical inference and (ii) what the target of statistical inference is.

4. This framework seems most relevant for explaining the uncertainty of either a continuous outcome or a multi-class outcome. For binary outcomes, the mean-variance relationship suggests to me that the results from using more typical Shapley values will be highly similar to those using this new definition. However, many of the experiments in Section 6 deal with binary outcomes. First, the authors should compare their results to those from using Shapley values for explaining the conditional mean. Second, the author should consider additional experiments dealing with other types of outcomes, to really highlight the utility of their proposed method.

**Questions:**

N/A

**Limitations:**

The authors discuss some limitations of their method.

---

> ### Author Rebuttal · Authors · 2023-08-10
>
> We thank TWY9 (henceforth R4) for their close reading and insightful feedback.
> We respond to specific issues below.
>
> W1. We thank R4 for pointing us to Williamson & Feng (2020), a reference we had not previously come across. Their SPVIM method is an elegant frequentist alternative to the Bayesian procedure we cite in the original manuscript. However, we dispute the claim that “a number of other papers…have studied how to explain the conditional variance of Y given X using Shapley values”–we are unaware of any such works, and would appreciate the reference(s) that R4 has in mind. The cited W&F paper, for instance, does *not* explicitly do this. Rather, W&F provide a procedure for performing inference using various global measures of “predictiveness”. Though their method could in principle be applied to games that model conditional variance or information theoretic quantities, the authors do not study or implement any such examples.
>
> R4 is correct to point out that existing inference procedures could be applied to our information theoretic Shapley values, a point we also acknowledge in Sect. 5. However, this is orthogonal to our main goals in this paper. First, we propose a number of novel information theoretic Shapley values, which we motivate by their ability to describe conditional dependencies beyond the first moment and their applicability in settings where labels are unavailable. Second, we show that conformal inference can be used to test whether the set of local Shapley values for a single feature tends to concentrate around zero. This is different from testing whether a single local or global feature attribution is significantly far from zero. Of course, one could in principle evaluate global importance by running $n$ tests using a local inference procedure, or compute a global measure upfront instead of aggregating local ones. The former is inefficient by comparison, while the latter requires a global value function, as opposed to the local alternatives we consider here. We have amended the manuscript to better contextualize our contribution and acknowledge the work of W&F.
>
> W2. R4 objects that Thm. 4.5 is “obvious”. We are somewhat sympathetic to this complaint with regards to result (a), which we include primarily for completeness. As for results (b) and (c), we must respectfully disagree with R4’s judgment. That context-specific independence (CSI) is sufficient but not necessary for zero marginal payoff, and that counterexamples to the necessity claim under CSI are measure zero, came to us as something of a surprise. These results rely on carefully constructed examples and an algebraic lemma from the 1970s. If these points were immediately obvious to R4 upon reading the definitions, then we commend R4 for their intuitive command of information and measure theory. We have amended the abstract to avoid all talk of “deep connections”, but are comfortable with the claim in Sect. 4 that our results establish a “somewhat subtle link between conditional independencies and information theoretic Shapley values.”
>
> W3. We thank R4 for pressing us to better motivate our inference procedure in Thm. 5.1 and to clarify our statistical target. First, we reiterate that our primary goal is to define and study a number of local attribution methods based on information theoretic value functions. As a secondary benefit, we may aggregate local attributions to test whether Shapley values concentrate around zero, which is evidence of a globally uninformative predictor. This is useful and efficient when local attributions have already been computed. However, if a researcher’s only goal is to test global importance, then they are better off using global measures upfront.
>
> Our statistical targets in Thm. 5.1 are the upper and lower extrema of *the oracle band*. Let $P_j$ be the true distribution of Shapley values for feature $j$, i.e. $\phi(j, \mathbf{x}) \sim P_j$, with corresponding quantile function $Q_j$. Then the oracle band for level $\alpha$ is defined as $C^*_j(\alpha) := \big[Q_j(\alpha / 2), Q_j(1 - \alpha / 2)\big]$. This band is *optimal* in the sense that it has the shortest length among all bands with valid coverage. Our previous version of this theorem relied on an implicit symmetry assumption to compute split conformal bands. Though symmetry holds for uninformative features, it may not hold in general and is inessential in practice. A more general solution is to report the empirical quantiles as estimated on $\mathcal{I}_2$. Our modified result therefore reads: $\mathbb{P} \big( \phi(j, \mathbf{x}^{(n+1)}) \in \big[\hat{Q}_j(\alpha / 2), \hat{Q}_j(1 - \alpha / 2)\big]  \big) \geq 1 - \alpha$. We have updated our coverage experiments accordingly (PDF, Table 1).
>
> R4 inquires about sources of error for these estimates. We identify three: (i) estimating the target function $h$ from finite samples; (ii) sampling values for out-of-coalition features; and (iii) sampling coalitions. Convergence rates as a function of (i) and (ii) are entirely dependent on the selected subroutines. With consistent methods for both, conformal prediction bands are provably close to the oracle band (see Lei et al., 2018, Thm. 8). We have conducted new experiments to empirically evaluate consistency under a range of conditions (see PDF, Fig. 1). As for (iii), we defer to the results of W&F, who show that with efficient estimators for (i) and (ii), and an extra condition on the minimum number of subsets, sampling $m = \Theta(n)$ coalitions is asymptotically optimal, up to a constant factor.
>
> W4. We thank R4 for the suggestion to compare our results against those of standard Shapley values to better understand how the two relate (PDF, Fig. 2). As expected, standard Shapley values for binary classification problems are highly predictive of information theoretic Shapley values. The relationship is somewhat less salient in the multiclass setting, as illustrated by our 10-class MNIST experiment (PDF, Fig. 2B).

---

> > ### Author Response · Authors · 2023-08-15
> > **Re: Rebuttal**
> >
> > Please let us know if there is anything more we can do to address reviewer comments or concerns. If we have resolved all major issues raised in the initial review, then we invite R4 to revise their score upward :)

---

> > > ### Comment · Reviewer_TWY9 · 2023-08-18
> > > **Thank you for the response**
> > >
> > > I thank the authors for their response. Couple of comments:
> > > 1. Re other methods that could be used for VI for conditional variance:
> > > * W&F is a general framework that can be instantiated with various definitions for predictiveness. One such value function could be the average squared error of the oracle model with respect to a variable subset, which corresponds to the average variance for the conditional distribution of Y given X_s. The VI measures in this paper can be defined for subsets of the population, leading to conditional variable importance measures.
> > > * Also, can one not apply the standard Shapley measure used in the literature, but to a ML algorithm that estimates the conditional variance as opposed to the conditional mean? Although the standard procedure doesn't have uncertainty quantification, it seems like it can be applied to estimate similar quantities.
> > >
> > > 2. I thank the authors for clarifying which parts of Theorem 4.5 are supposed to be original. Perhaps this can be clarified a bit more in the text. Yes, I agree part (c) is not that obvious, and the link to the faithfulness condition used in causal inference is interesting.
> > >
> > > 3. I thank the authors for clarifying the sources of uncertainty. As such, I would suggest revising the manuscript with these assumptions regarding consistency and coverage of the Shapley values is in fact asymptotic.
> > >
> > > I will increase my score by 1.

---

> > > > ### Author Response · Authors · 2023-08-18
> > > > **Re: Reviewer comment**
> > > >
> > > > Many thanks to R4 for taking the time to read through our rebuttal and revise their score upward. In reply to R4's comments:
> > > >
> > > > 1. We agree that W&F's framework (or, for that matter, the Slack et al. Bayesian method we originally cited) could be applied to the value functions we describe – indeed, they can be applied to *any* value function. Our novelty lies primarily in proposing and studying a number of information theoretic value functions that are not in widespread use, and illustrating their applicability for settings that require uncertainty quantification (e.g., active learning and out-of-distribution detection). In addition, we outline a conformal inference procedure for testing whether Shapley values for a given feature concentrate around zero, a task that is subtly different from both BayesSHAP (which is only for local inference, and would therefore require $n$ tests to draw a global inference) and SPVIM (which requires a global value function upfront, as opposed to the local measures we aggregate). As for R4's question about building a model to estimate conditional variance, the answer is yes – that is precisely what our $h$ function does, although details vary depending on whether we aim to explain the epistemic or aleatoric uncertainty (see Sect. 5). For examples of how this works in practice, see Sect. 6.
> > > >
> > > > 2. We thank R4 for reconsidering their position on Thm. 4.5, especially with regard to item (c). We also find the faithfulness connection to be quite interesting! We will be sure to clarify the relative significance of each item in the text, and will try to move some of the discussion from Appx. A to the main text if space allows.
> > > >
> > > > 3. R4's point on asymptotic consistency and coverage in Thm. 5.1 is well-taken. We have thoroughly revised the text to address R4's questions on this topic, including an expanded discussion on sources of uncertainty when estimating the oracle band. We thank R4 again for challenging us to work through this result more carefully. The manuscript is much improved following this revision.
> > > >
> > > > We reiterate our appreciation for R4's thoughtful review and reply. We would be happy to address any further questions or comments R4 may have.

---

### Official Review · Reviewer_yQq8 · 2023-07-03

**Soundness:** 3 good
**Presentation:** 3 good
**Contribution:** 3 good
**Rating:** 6
**Confidence:** 3

**Summary:**

I have previously reviewed this paper at another conference in which, after reviewers' discussion, it was a borderline reject, below my review is enriched with the previous conference discussions.

This work proposes calculating a modified Shapley value where the coalitional game represents the entropy of the predictive distribution.
A high-scoring feature will increase the entropy, and a low-score feature will decrease it.

Sections 4 and 5 describe the properties of their value function, along with a couple of alternative value functions. They discuss how their value function connects to the notion of information gain, a decomposition of uncertainty into aleatoric and epistemic sources, and a PAC-like technique for testing if a feature's mean Shapley value is near zero.

Various experiments are performed to illustrate use cases and validate correctness.

**Strengths:**

- The idea of using KL divergence and Entropy in combination with Shapley Values to attempt to quantify the individual importance of features is novel and potentially very useful. Not many papers have delved into the topic of attributing uncertainty to input features.

 - Nice job deriving properties from the two main coalitional games they discussed

- Extensive set of tests given.

**Weaknesses:**

In the final discussion, the main reasons to reject were the following:

 - Developing metrics to verify the correctness of their results
 - Making their implementation consistent with the methods section.
 - Experimental Limitations: the experimental design is based on papers from Computer Vision, where pure covariate shift is identifiable.
 - Some missing discussions about related work

**Questions:**

Since the paper was previously rejected and no major modifications have happened. Why would the authors expect the paper to be accepted this time?

**Limitations:**

-

---

> ### Author Rebuttal · Authors · 2023-08-09
>
> We thank reviewer yQq8 (henceforth R3) for taking the time to read and comment on our manuscript (again!). We learned a great deal during the review process for ICML, incorporating reviewer feedback to improve the manuscript in numerous ways—an improvement widely acknowledged by reviewers, who revised their scores upward by 2-3 points on average. We strenuously object to the charge that “no major modifications have happened” since that submission. In fact, Sect. 4 was rewritten in its entirety, with major updates to our theoretical results (including the introduction of context-specific independence structures that went unmentioned in the previous manuscript). We added numerous references to our literature review, including further discussion on related methods such as LossSHAP and SAGE, and amended the experiments to more clearly implement our theory, computing explanations for both epistemic and aleatoric uncertainty under a range of function classes and sampling procedures, in addition to our recently added convergence experiments. We hope R3 will agree that this manuscript has come a long way since we submitted an initial draft to ICML some six months ago.
>
> We reply to R3’s critiques as laid out in the “Weaknesses” section below:
>
> W1. We appreciate the suggestion, also made by other reviewers, to include more quantitative experiments to evaluate the soundness of our method. Toward that end, we have conducted a simulation experiment in which ground truth Shapley values can be calculated in closed form to see how our estimation procedures fare under a range of data generating processes and imputation methods for out-of-coalition feature values. Echoing results from other XAI studies (Olsen et al., 2023), we find that no single method dominates throughout. However, we tend to converge on true information theoretic Shapley values when sample sizes are large and dependencies between features are well modeled. See our comment to all reviewers above for more details.
>
> W2. Better aligning our methods and experiments section is one of the major changes we have made to the manuscript since the initial ICML submission. This includes expanding the experiments to cover cases of both epistemic and aleatoric uncertainty (as opposed to just total uncertainty), and comparing different methods for reference distribution sampling across different base algorithms. We would be open to adding new experiments to better align these sections if R3 has some specific suggestion(s) in mind?
>
> W3. We are somewhat confused by the claim that “the experimental design is based on papers from Computer Vision.” Sect. 6 includes just a single image data example—from the canonical MNIST—in addition to examples from natural language processing (Fig. 1B), and numerous tabular datasets (Figs. 2, 3). We have made a deliberate effort to include a diverse set of experiments spanning structured and unstructured data types, as well as classification and regression examples. We fail to see how our experimental design is limited to computer vision applications.
>
> W4. We have attempted to cover a wide body of literature in this manuscript, including papers on explainable AI, uncertainty quantification, information theory, and application areas such as active learning, covariate shift detection, feature selection, and classification with reject option. R2 specifically complimented our “thorough literature review”. Our bibliography spans some 80+ references, compared to 55 references in the earlier version of our work that was reviewed at ICML. In any event, we are happy to expand Sect. 2 or Sect. 7 with extra references if R3 has any specific titles in mind.
>
> We hope this rebuttal goes some way toward addressing R3’s concerns, and kindly request that they consider revising upward :)

---

> > ### Author Response · Authors · 2023-08-15
> > **Re: Rebuttal**
> >
> > Please let us know if there is anything more we can do to address reviewer comments or concerns. If we have resolved all major issues raised in the initial review, then we invite R3 to revise their score upward :)

---

> > > ### Comment · Reviewer_yQq8 · 2023-08-16
> > >
> > > Many thanks! I see how the authors addressed the questions and what has changed in the submissions.
> > >
> > > I find the paper in a good level of maturity. I will consider increasing my score after the discussion with other reviewers.

---

### Official Review · Reviewer_DQBN · 2023-07-06

**Soundness:** 3 good
**Presentation:** 3 good
**Contribution:** 2 fair
**Rating:** 4
**Confidence:** 4

**Summary:**

This paper introduces a method for explaining the uncertainty in predictions made by DNNs. The authors extend the Shapley values from explaining the value of DNN outputs to explaining the uncertainty of the DNN outputs. Then, the authors demonstrate the close relationship between these Shapley values and the conditional independencies between inputs and outputs. Additionally, the authors provide a theoretical bound to support the coverage of the explanation obtained by the proposed method.

**Strengths:**

1.	The paper is well-organized and easy to follow. The authors also provide a thorough literature review.
2.	The authors prove some propositions and theorems to support the reliability of the proposed definition of Shapley values in DNNs.
3.	It is good that the authors recognize the presence of both epistemic and aleatoric uncertainty in the prediction of DNNs.


**Weaknesses:**

1.	The motivation behind investigating the Shapley values of inputs towards the uncertainty of predictions is not convincing. While the proposed information-theoretic Shapley value reveals which variables contribute to prediction uncertainty, it fails to explain which variables are responsible for correct/incorrect predictions. The original Shapley values defined on the model output can provide this information. On the other hand, the results in Figure 1 do not demonstrate the superiority of the proposed information-theoretic Shapley value. This is because the features that increase predictive uncertainty can also be considered as pixels with negative Shapley values toward the correct prediction. Therefore, the information-theoretic Shapley value does not offer significant additional insights or information.
2.	How should I understand the Shapley values defined on the value functions $v_{KL}$ and $v_{CE}$? When using these value functions, the output $v(N)$ is always 0.
3.	Is the proposed information-theoretic Shapley value a local explanation for individual samples or a global explanation over all training samples? If it is a local explanation, what does the coverage of the explanation represent? If it is a global explanation, how is the value function defined over different samples?
4.	In Figure 2, many perturbed inputs are assigned negative Shapley values for predictive uncertainty, which is perplexing. In my understanding, a negative contribution to the predictive uncertainty indicates that this feature is discriminative. However, the perturbed features are typically expected to be non-discriminative.
5.	There is a lack of quantitative experiments to validate the accuracy of the proposed Shapley values in estimating the uncertainty/variance of the prediction. Figure 3(A) just provides qualitative results and Figure 3(B) focuses solely on the ranking of features. I suggest the authors conduct a quantitative experiment to show that the proposed method could precisely explain and estimate the variance of the prediction.


**Questions:**

1.	Further discussion is needed to clarify the motivation and advantages of explaining predictive uncertainty.
2.	It is suggested that the authors perform a quantitative experiment to demonstrate the ability of the proposed method to accurately explain and estimate prediction variance.


**Limitations:**

The authors have discussed the limitation of the method.

---

> ### Author Rebuttal · Authors · 2023-08-09
>
> We thank reviewer DQBN (henceforth R2) for their time and feedback. We were pleased to see that R2 found our paper “well-organized and easy to follow”, but would like to clarify several points of potential confusion that were raised in the “Weaknesses” section.
>
> W1. R2 objects that our method “fails to explain which variables are responsible for correct/incorrect predictions.” This is true – and by design. R2 appears to be conflating “uncertainty” (our target) with “error” (not our target). A method such as LossSHAP (Lundberg et al., 2020), which seems closer to what R2 has in mind, would explain the pointwise loss, e.g. the squared residual of each prediction in a regression example. However, this requires labels. As noted throughout the manuscript, *our method is unique in handling cases where labels are unavailable*. This can occur, for instance, in clinical medicine or online advertising, where outcomes are delayed or expensive to collect. As we point out in Sect. 4, there is a close relationship between our proposed games and LossSHAP (using likelihood-based loss functions). Specifically, our games represent average LossSHAP values as we marginalize over $Y$ with respect to different conditional distributions. This is the best we can do without access to labels. Our experiments in covariate shift (Sect. 6.2), which can also be used to explain the selections of an acquisition function for an active learning algorithm, illustrate the utility of our information theoretic Shapley values in cases where LossSHAP is impossible to compute.
>
> Beyond simply removing the need for labels, a key motivation for explaining uncertainty rather than error is that throughout many areas in machine learning there exist methods that are based on uncertainty, and we may need to explain their behavior. A few examples:
> Many active learning algorithms select their queries based on (different notions of) uncertainty, and one may want to know “why did our active learning algorithm query those instances?”
> Many out of distribution (OOD) detection algorithms make their OOD assessments based on some notion of uncertainty, and one may want to know “why did our OOD method assess this instance to be out of distribution?”
> Our methods are uniquely well-suited to handle such questions.
>
> W2. We are unsure why R2 is under the impression that $v_{KL}$ and $v_{CE}$ are invariant functions that map all inputs to zero. This is false. Quoting from Sect. 4: $v_{KL}(S, \mathbf{x})$ “can be interpreted as $-1$ times the excess number of bits one would need on average to describe samples from $Y \mid \mathbf{x}$ given code optimized for $Y \mid \mathbf{x}^S$.” The cross entropy value function has a similar interpretation, and is indeed equivalent up to an additive constant. In other words, these value functions measure the information gap between two conditional distributions for $Y$: one based on the complete vector $\mathbf{x}$, and another based on the partial vector $\mathbf{x}^S$. Large values can occur when informative features are excluded from $S$. To take an extreme example, consider a case in which $H(Y \mid X) = 0$, $H(Y) > 0$, and $S = \emptyset$. Then for all $\mathbf{x}$, we have $v_{KL}(S, \mathbf{x}) = v_{CE}(S, \mathbf{x}) = H(Y)$. In general, these value functions can be made arbitrarily large by letting $Y$’s prior entropy tend toward infinity while its local posterior entropy tends toward zero.
>
> W3. Our proposed value functions compute exclusively *local* Shapley values, as we emphasize in several places throughout the manuscript, e.g. when contrasting against Covert et al.’s (2020) SAGE method in Sects. 2 and 4. However, we acknowledge that this may complicate the interpretation of Thm. 5.1 without further elaboration. Our coverage guarantee pertains not to an individual Shapley value $\phi(j, \mathbf{x})$ but rather to the set of all Shapley values for a single feature $X_j$. As we note in the manuscript, the motivation for bounding the spread of this set is that “These results can inform decisions about feature selection, since narrow intervals around zero are necessary (but not sufficient) evidence of uninformative predictors.” In other words, we aggregate local attributions to draw a global inference. This connects with the conditional independence results of Thm. 4.5, which license an interpretation of the marginal payoff function as a conditional dependence measure. We have updated Sect. 5 to make this point more explicit. See also our reply to R4, who raises some questions about this.
>
> W4. Negative Shapley values in this game correspond to feature values that decrease predictive uncertainty relative to a data-dependent baseline (i.e., the prior entropy $H(Y)$). Extremely low values following perturbation indicate over-confidence in the corresponding prediction, a common case of miscalibration under covariate shift (see, e.g., Ovadia et al., 2019). Identifying the sources of such miscalibration can inform decision making. We have added some text to Sect. 6.2 to clarify this point.
>
> W5. We appreciate R2’s suggestion to include a quantitative experiment to demonstrate our method’s ability to accurately explain predictive uncertainty. We have conducted a simulation in which this quantity can be calculated in closed form, and estimated the corresponding Shapley values using a range of different pipelines for imputing out-of-coalition feature values. See the comment to all reviewers above for more details on this experiment.
>
> References:
>
> -Lundberg et al., 2020: https://www.nature.com/articles/s42256-019-0138-9
>
> -Covert et al., 2020: https://proceedings.neurips.cc/paper/2020/file/c7bf0b7c1a86d5eb3be2c722cf2cf746-Paper.pdf
>
> -Ovadia et al., 2019: https://proceedings.neurips.cc/paper_files/paper/2019/file/8558cb408c1d76621371888657d2eb1d-Paper.pdf

---

> > ### Comment · Reviewer_DQBN · 2023-08-14
> >
> > I would like to thank the authors for their response, but some concerns still remain.
> >
> > Weakness 1. I'm clear about the difference between explaining uncertainty and explaining the error/model output. My concern is that explaining the model output usually provides more information than just explaining the uncertainty. Although there are some cases where the label is unknown, explaining the output value of each category may also help people understand the inference of the model.
> >
> > Weakness 2. When taking $v_{KL}$ and $v_{CE}$ as the value function, the output $v(N)$ given all input features is always zero, because it computes the distance between two same distributions ($p(y|x)$). Thus, my question is, what does the Shapley value in this scenario mean? Using the traditional reward function (maybe the model output or loss), the Shapley value refers to how much contribution each input feature has to the output score/loss. But with $v_{KL}$ and $v_{CE}$ as the value function, I don't know how to understand the Shapley value.
> >
> > Weakness 4. What does the miscalibration refer to? The authors did not answer why the randomly perturbed features were explained as discriminative features (leading to the over-confidence of the prediction).

---

> > > ### Author Response · Authors · 2023-08-14
> > > **Re: Official Comment by Reviewer DQBN**
> > >
> > > We thank R2 for taking the time to reply to our rebuttal. We respond to their new comments below.
> > >
> > > W1. Explaining individual predictions provides information at a different level of detail. Existing tools are already well-suited to these applications. However, as we note in our rebuttal, current methods may not be useful in other areas that rely on uncertainty evaluation such as active learning and OOD detection.
> > > Though one could in principle inspect SHAP explanations for each candidate label to probe for sources of uncertainty, this becomes infeasible with more than two or three classes and is unnecessarily indirect.
> > > Moreover, focusing on the first conditional moment ignores higher order information, as we explain in our motivating example at the top of Sect. 4 where $X, Z \sim \mathcal{U}(0, 1)^2$ and $Y \sim \mathcal{N}(X, Z^2)$. Standard SHAP values will assign zero importance to $Z$ here, although the variable does in fact provide information about the conditional distribution of $Y$. In summary, our proposed Shapley values *complement* rather than *supplant* existing alternatives.
> > >
> > > W2. Apologies for misunderstanding R2’s original comment – we see now that what R2 calls $N$ is what we call $[d]$.
> > > Often in Shapley style games, the value function has positive range with $v(\emptyset) = 0$. However, as R2 correctly observes, zero is the *ceiling* rather than the *floor* for $v_{KL}$ and $v_{CE}$. That is, these value functions have *negative* range, with $v_{KL}(\emptyset, \mathbf{x})$ representing the negative KL-divergence between prior and posterior distributions for $Y$.
> > > As we explain in Prop. 4.1 and the surrounding text, resulting marginal payouts $\Delta_{KL}(S, j, \mathbf{x})$ represent the information gained by $X_j = x_j$ when we already know feature values for coalition $S$, assuming the true target distribution is $Y \mid \mathbf{x}$.
> > > As we explain in Prop 4.2 and the surrounding text, resulting Shapley values represent the contribution in bits to the KL-divergence between prior and posterior distributions.
> > >
> > > W4. Miscalibration in this context means that the model is over-confident in its prediction. Perturbing a feature can lead to overconfidence if, for instance, the model infers a monotonic relationship between $X$ and $P(Y \mid X)$ during training and then sees an unusually high value for $X$ at test time.
> > > If random perturbations decrease predictive uncertainty, we infer that (a) the feature in question must be one the model relies on for making predictions, otherwise we would see no change in output; and (b) the model is extrapolating with too much confidence to anomalous regions of the feature space.

---

> > > > ### Author Response · Authors · 2023-08-15
> > > > **Re: Official Comment**
> > > >
> > > > Thanks again to R2 for taking the time to go through our rebuttal. Please let us know if there is anything more we can do to address reviewer comments or concerns. If we have resolved all major issues raised in the initial review, then we invite R2 to revise their score upward :)

---

### Official Review · Reviewer_2QPT · 2023-07-08

**Soundness:** 3 good
**Presentation:** 4 excellent
**Contribution:** 3 good
**Rating:** 7
**Confidence:** 3

**Summary:**

Authors aim to explain uncertainty in model outputs by adapting Shapely value framework. In specific, authors try to explain predictive distributions through the lens of entropy, cross-entropy, KL and information gain. Authors offer some theoretic interpretation of the Shapley values adapted with these metrics and, through experiments, show the efficacy of the entropy based variant of Shapely values.

**Strengths:**

1. I would like to thank the authors for presenting their work in a very thoughtful manner. I find most of the paper to be clear and intuition which follows propositions and theorems to be very helpful. I like the how the paper introduces theory to offer intuition and make some simplifications to make the method more practical.

2. With raising importance in uncertainty quantification in ML, the work presented in this paper is very relevant to the community.

3. The paper not only presents theoretical analysis of the modified variants of the Shapley values but also offers empirical evidence of how their proposed method works beyond the assumptions made in the theoretical analysis.

**Weaknesses:**


I might be missing something, but I am unable to understand the specific details of the experiments by reading the main paper and the appendix. (even the supplementary code didn't have much documentation). For example, I couldn't find the details of ensembles method used. What is the size of ensemble (`B` in Sec 5)? Which type of ensemble [5] is being used? What was the batch size how many training epochs were run?

I think there is a line of work on uncertainty estimation and it's interpretation which is missing and might be useful. (A subset of these papers might be useful: [1] , [2], [3], [4])


[1] From Predictions to Decisions: The Importance of Joint Predictive Distributions (https://arxiv.org/pdf/2107.09224.pdf)
[2] Epistemic neural networks (https://arxiv.org/pdf/2107.08924.pdf)
[3] Evaluating High-Order Predictive Distributions in Deep Learning (https://proceedings.mlr.press/v180/osband22a/osband22a.pdf)
[4] Marginal and Joint Cross-Entropies & Predictives for Online Bayesian Inference, Active Learning, and Active Sampling (https://arxiv.org/pdf/2205.08766.pdf)
[5] Ensembles for Uncertainty Estimation: Benefits of Prior Functions and Bootstrapping (https://arxiv.org/pdf/2206.03633.pdf)

**Questions:**

1. In Section 6, even through the problems were explained in some detail, I find Section 6 and Appendix C to be lacking in details of methods used and their hyperparameters. Can you please include these details?

2. Some parts of the paper refers to active-learning experiments in the experiments section, but it was not obvious to me how the experiments in the Section 6 are related to active learning. Can you clarify this?

3. In Section 5, authors present computation of entropy (h_t) with a single point estimate and then computation of aleatoric entropy (h_a) with an ensemble of particles. It is not clear how h_t is estimated when we use an ensemble of particles. Can you clarify this?

4. There is reference to script D in line 219. I don't think this was introduced before or referenced later. It would be useful to replace this with an appropriate term.

**Limitations:**

Please refer to the weakness section.

I am willing to increase my score if authors provide more details about their experiments either in the main paper or the appendix.

---

> ### Author Rebuttal · Authors · 2023-08-09
>
> We sincerely thank reviewer 2QPT (henceforth R1) for their attentive comments and overall positive assessment. We were especially pleased to find that R1 considered our presentation “very thoughtful” and judged “the paper to be clear and intuition which follows propositions and theorems to be very helpful.” We appreciate R1’s suggestion to review other methods for computing and interpreting predictive uncertainty in machine learning. We have incorporated these references into Sect. 2 (Related Work) and Sect. 7 (Discussion).
>
> R1 raises several important questions, which we address below.
>
> Q1. We agree that the presentation in Sect. 6 and Appx. C could be more thorough, and have amended the text to include previously excluded details. In particular, for our image data example, we train a neural network with the following model architecture: (1) A convolutional layer with 10 filters of size 5x5, followed by max pooling of size 2x2, ReLU activation, and a dropout layer with probability 0.3. (2) A convolutional layer with 20 filters of size 5x5, followed by a dropout layer, max pooling of size 2x2, ReLU activation, and a dropout layer with probability 0.3. (3) Fully connected (dense) layer with 320 input features and 50 output units, followed by ReLU activation and a dropout layer. (4) Fully connected layer with 50 input features and 10 output units, followed by softmax activation. We train with a batch size of 128 for 20 epochs at a learning rate of 0.01 and momentum 0.5. For Monte Carlo dropout, we do 50 forward passes to sample $B = 50$ subnetworks. This information has now been added to Appx. C.2. The XGBoost ensembles vary in size from $B = 50$ to $B = 100$, with details provided in Sect. 6.
>
> Q2. We thank R1 for pointing out that it may not be immediately clear how our experiments in Sect. 6 connect to the active learning (AL) setting we mention at several points throughout the text. The connection is strongest in Sect. 6.2 on covariate shift. Following several other authors (e.g., Sugiyama et al., 2007; Quiñonero-Candela et al., 2009), we see this task as basically continuous with AL, since an effective acquisition function will tend to prioritize anomalous samples that appear to be drawn from a different distribution than the training data. These will naturally have high epistemic uncertainty. It is common in the AL literature to use epistemic uncertainty to select for which instances to query the label. This is inherently the case in Query-by-Committee based methods, and has recently become more common also in the uncertainty sampling literature (e.g., Nguyen et al., 2022). We have rewritten the text in this subsection to clarify the connection between these tasks.
>
> Q3. We thank R1 for observing that our discussion in Sect. 5 is potentially ambiguous with respect to the role of individual basis functions $f^b, b \in [B]$ in defining the total entropy $h_t$. This depends on how we construct the ensemble predictor $f_y(\mathbf{x}) := p(y \mid \mathbf{x})$. The simplest method—widely used in, e.g., random forests and deep ensembles—is to average over the basis functions: $f_y(\mathbf{x}) := B^{-1} \sum_{b=1}^B f^b(\mathbf{x})$. By contrast, in boosting models, predictions represent the sum (rather than the average) of basis model outputs. For more details on aggregating basis functions for probabilistic predictions in random forests, see (Malley et al., 2012); for more on deep ensembles, see (Lakshminarayanan et al., 2017); and for details on uncertainty quantification in gradient boosting, see (Malinin et al., 2021). We have expanded the text in Sect. 5 to include these references and resolve the ambiguity identified by R1.
>
> Q4. The reference distribution $\mathcal{D}$ was introduced in Sect. 3, specifically the subsection on Shapley values. We have amended the text in Sect. 5 to remind readers of this connection.
>
> We hope our comments have resolved all of R1’s concerns. If so, we invite them to revise their score upward :)
>
> References:
>
> -Sugiyama et al., 2007: https://jmlr.org/papers/v8/sugiyama07a.html
>
> -Quiñonero-Candela et al., 2009: https://mitpress.mit.edu/9780262545877/dataset-shift-in-machine-learning/
>
> -Malley et al., 2012: https://www.ncbi.nlm.nih.gov/pmc/articles/PMC3250568/
>
> -Lakshminarayanan et al., 2017: https://proceedings.neurips.cc/paper_files/paper/2017/file/9ef2ed4b7fd2c810847ffa5fa85bce38-Paper.pdf
>
> -Malinin et al., 2021: https://openreview.net/pdf?id=1Jv6b0Zq3qi
>
> -Nguyen et al. 2022: https://link.springer.com/article/10.1007/s10994-021-06003-9

---

> > ### Comment · Reviewer_2QPT · 2023-08-14
> >
> > I would like to thank the authors for addressing the points I raised. As mentioned in my review, I am increasing the score to accept.

---

### Author Rebuttal · Authors · 2023-08-09

We thank all the reviewers for their thoughtful comments and constructive feedback. Working through our replies has deepened our understanding of the material and helped us further refine our manuscript.

We reply to all individual reviewer comments below. However, we open with one general note here, since it came up in multiple reviews. We appreciate the suggestion to include a quantitative experiment that demonstrates our method’s ability to accurately explain predictive uncertainty. We have designed a simulation experiment, loosely inspired by Aas et al. (2021), in which Shapley values under the entropy game have a closed form solution, and tested a range of methods for estimating these quantities in regression examples (see attached PDF, Fig. 1). As with previous work on Shapley value computation, the main considerations to keep in mind here are (i) the accuracy of the conditional entropy estimator (analogous to the base model in more classic XAI settings); (ii) the accuracy of the imputation method for estimating out-of-coalition feature values; and (iii) the coalition sampler. For this experiment, we simulate multivariate normal data with $d=4$ predictors and a Toeplitz covariance matrix with variable autocorrelation. The conditional log-variance of $Y$ is given by a linear function of $\mathbf{X}$. We estimate this conditional variance using multiple regression on the log squared residuals of a linear model for $\mathbb{E}[Y \mid \mathbf{X}]$, and exhaustively enumerate candidate coalitions (generally feasible with $d$ on the order of ~10). For imputation schemes, we compare KernelSHAP, maximum likelihood, copula methods, empirical samplers, and ctree (see Olsen et al., 2023 for definitions of each and a benchmark study of conditional sampling strategies for Shapley values). We note that this is strictly more difficult than the standard case, since $\text{Var}[Y \mid \mathbf{X}]$ is not directly observed but must be estimated from the data. No single method dominates throughout, but we show that under some combinations, we are able to converge on the true information theoretic Shapley value with existing imputation pipelines. This demonstrates both the accuracy of our method and its modularity with respect to reference distribution samplers. We thank the reviewers for pressing us on this point and encouraging us to add this valuable experiment to Sect. 6.

The attached PDF also includes some revised or expanded experiments, in reply to particular reviewer comments (see below).


References:

-Aas et al., 2021: https://www.sciencedirect.com/science/article/pii/S0004370221000539

-Olsen et al., 2023: https://arxiv.org/abs/2305.09536

---

> ### Author Response · Authors · 2023-08-20
>
> We would like to once again thank all the reviewers for their contributions before the discussion period comes to a close. If there are any further comments or questions we can address, please let us know and we will do our best to reply promptly. We are very grateful for the helpful feedback and stimulating discussion!

---

### Decision · Program_Chairs · 2023-09-21

**Decision:**

Accept (poster)

**Comment:**

This paper is concerned with explaining the uncertainty of model outputs. The novelty lies primarily in proposing and studying a number of information theoretic value functions that are not in widespread use, and illustrating their applicability for settings that require uncertainty quantification (e.g., active learning and out-of-distribution detection). All reviewers agreed with this contribution and most found this useful. As the authors indicated the proposed Shapley values complement rather than supplant existing alternatives. With the raising importance of uncertainty quantification in ML, this work is very relevant to the community. The paper also provide a non-trivial theoretical analysis as acknowledge by the most critical reviewer. Finally, the authors provided a clear and solid rebuttal addressing the concerns raised by the reviewers. I would encourage the authors to incorporate the feedback to further strengthen the paper.

It should be noted that the paper was previously rejected by ICML. All pending issues have been taken into account and the work has been significantly improved (and expanded as requested).